# ImageNet-OOD: Deciphering Modern Out-of-Distribution Detection Algorithms

**William Yang**[*]**, Byron Zhang**[*]**, Olga Russakovsky**
Department of Computer Science, Princeton University, Princeton, NJ, USA
{williamyang,zishuoz,olgarus}@cs.princeton.edu

## Abstract

The task of out-of-distribution (OOD) detection is notoriously ill-defined. Earlier works focused on new-class detection, aiming to identify label-altering data distribution shifts, also known as "semantic shift." However, recent works argue for a focus on failure detection, expanding the OOD evaluation framework to account for label-preserving data distribution shifts, also known as "covariate shift." Intriguingly, under this new framework, complex OOD detectors that were previously considered state-of-the-art now perform similarly to, or even worse than, the simple maximum softmax probability baseline. This raises the question: what are the latest OOD detectors actually detecting? Deciphering the behavior of OOD detection algorithms requires evaluation datasets that decouple semantic shift and covariate shift. To aid our investigations, we present ImageNet-OOD, a clean semantic shift dataset that minimizes the interference of covariate shift. Through comprehensive experiments, we show that OOD detectors are more sensitive to covariate shift than to semantic shift, and the benefits of recent OOD detection algorithms on semantic shift detection is minimal. Our dataset and analyses provide important insights for guiding the design of future OOD detectors.[1]

## 1 Introduction

Out-of-distribution (OOD) detection aims to identify test examples sampled from a different distribution than the training distribution. In the context of computer vision, an OOD detector is simply a post-hoc score calibration function that operates on a trained image classification model. Previous work has proposed tackling this problem from two perspectives: new-class detection and failure detection. In new-class detection, OOD detectors are expected to identify new object categories for the purpose of data collection and continual learning (Liu et al., 2020; Hendrycks et al., 2022; Wang et al., 2022; Liang et al., 2018; Kim et al., 2022). Recent works have motioned to shift the objective from new-class detection to the scenario of failure detection, where OOD detectors are expected to identify misclassified examples to promote the safety and reliability of deep learning models in real-world applications (Jaeger et al., 2023; Zhu et al., 2022; Averly & Chao, 2023; Guérin et al., 2023). However, evaluating OOD detectors via failure detection benchmarks yields one unanimous finding: no modern OOD detector surpasses the performance of the simple maximum softmax probability (MSP) (Hendrycks & Gimpel, 2017) baseline. In light of this drastic discrepancy, we need to take a step back to address the question: what are modern OOD detectors actually detecting?

Common literature in OOD detection separates distribution shifts into semantic (label-altering) and covariate (label-preserving) shifts (Hsu et al., 2020; Tian et al., 2021; Yang et al., 2021b). Understanding detection behavior for either type of shift requires proper evaluation datasets that decouple semantic shifts from covariate shifts (Yang et al., 2021a). Many OOD detection datasets (Wang et al., 2018; Hendrycks et al., 2021; Galil et al., 2023) set ImageNet-1K (Russakovsky et al., 2015) as in-distribution (ID) and subsets of ImageNet-21K (Deng et al., 2009) as out-of-distribution (OOD). Since ImageNet-1K is also a subset of ImageNet-21K, both datasets' class labels are derived from the WordNet (Fellbaum, 1998) hierarchy and the images share the same data collection process. However, these datasets often contain contamination from ID images, which violates one of the key

---

[*]Equal contribution.
[1] Code and data is at https://github.com/princetonvisualai/imagenetood

assumptions of OOD detection for both new-class and failure detection (Bitterwolf et al., 2023). Consequently, several human-annotated datasets have been constructed, though using data sources outside the ImageNet family (Hendrycks et al., 2022; Wang et al., 2022; Bitterwolf et al., 2023), introducing unforeseen covariate shifts due to changes in the data source and collection process.

In this paper, we design an OOD detection dataset that can accurately assess the impact of semantic shift without the influence of covariate shifts. Concretely, we introduce **ImageNet-OOD**, a clean, manually-curated, and diverse dataset containing 31,807 images from 637 classes for assessing semantic shift detection using ImageNet-1K as the ID dataset. ImageNet-OOD minimizes covariate shifts by curating images directly from ImageNet-21K while removing ID contamination from ImageNet-1K through human verification. We identify and remove multiple sources of semantic ambiguity arising from inaccurate hierarchical relations in ImageNet labels. Additionally, we remove images with visual ambiguities arising from the inconsistent data curation process of ImageNet. Using ImageNet-OOD and three ImageNet-1K-based covariate shift datasets, we perform extensive experiments on nine OOD detection algorithms across 13 network architectures, from both new-class detection and failure detection perspectives to make the following findings:

1. Modern OOD detection algorithms are even *more* susceptible towards detecting covariate shifts than semantic shift compared to the baseline MSP (Hendrycks & Gimpel, 2017).

2. In ImageNet-OOD, which exhibits only semantic and limited covariate shift, modern OOD detection algorithms yield very little improvement over the baseline on new-class detection.

3. Modern OOD detection algorithms only improve on previous benchmarks by ignoring incorrect ID examples rather than detecting OOD examples, causing performance disparity between the task of new-class detection and failure detection.

## 2 EXISTING DEFINITIONS AND FORMULATIONS

**Problem Setup.** For image classification, a dataset $D_{tr} = \{(x_i, y_i); x_i \in \mathcal{X}, y_i \in \mathcal{Y}\}$ sampled from training distribution $P_{tr}(x, y)$ is used to train some classifier $C : \mathcal{X} \rightarrow \mathcal{Y}$. In real-world deployments, distribution shift occurs when classifier $C$ receives data from test distribution $P_{te}(x, y)$ where $P_{tr}(x, y) \neq P_{te}(x, y)$ (Moreno-Torres et al., 2012). An OOD detector is a scoring function $s$ that maps an image $x$ to a real number $\mathbb{R}$ such that some threshold $\tau$ arrives at a detection rule $f$:

$$f(x) = \begin{cases} \text{in-distribution} & \text{if } s(x) \geq \tau \\ \text{out-of-distribution} & \text{if } s(x) < \tau \end{cases} \tag{1}$$

Common changes within the data distribution fall under two categories: covariate shift, which is label-preserving (i.e. concerns examples only from training classes), and semantic shift, which is label-altering (concerns examples only from new classes).

**Covariate Shift.** Covariate shift occurs in the test data when the marginal distribution with respect to the image differs from the training data: $P_{tr}(x) \neq P_{te}(x)$ (Yang et al., 2021b), while the label distribution remains fixed: $P_{tr}(y|x) = P_{te}(y|x)$. In this work, we will use three popular datasets with covariate shifts with respect to ImageNet-1K (Russakovsky et al., 2015): ImageNet-C (Hendrycks & Dietterich, 2019), ImageNet-R (Hendrycks et al., 2020), and ImageNet-Sketch (Wang et al., 2019).

**Semantic Shift.** Semantic shift occurs when a given set of semantic labels $Y_{tr} \subset \mathcal{Y}$ from the training distribution and semantic labels $Y_{te} \subset \mathcal{Y}$ from a test distribution have the following property: $Y_{tr} \cap Y_{te} = \emptyset$, such that $P_{tr}(y) = 0 \ \forall y \in Y_{te}$. To best study the effect of semantic shifts, we propose a new dataset, ImageNet-OOD, that minimizes the degree of covariate shifts.

**Evaluation.** Many OOD detection benchmark commonly use AUROC as a threshold free way of estimating detection performance by calculating the area under false positive rate vs. true positive rate curve, considering ID as positive and OOD as negative.

**Classical Approach: New-class Detection.** The majority of work in OOD detection has formulated the problem as new class-detection: the notion of ID is defined by the class labels of the data

source and therefore is model agnostic. In other words, an image $x$ is considered ID if it comes from a class $y \in Y_{tr}$ and OOD otherwise. OOD detection with this goal focuses exclusively on semantic shift: the detection of novel classes. Although the objective is analogous to that of Open-Set Recognition (Vaze et al., 2022), previous OOD detection works often motivated this goal to prevent model failures (Yang et al., 2021b). We provide more intricate discussion on the subtle differences in task formulation of previous OOD detection works in Section A of the Appendix.

**Modern Approach: Failure Detection.** Recent works argued to approach the OOD detection problem from the first principle. Instead of defining "ID" and "OOD" with regard to the data sources, the distinction is directly specified by the model's prediction. An image is ID if the model correctly classifies an image and OOD otherwise. These recent works converged on the conclusion that recent advances in OOD detection do not result in any improvement for the task of failure detection (Jaeger et al., 2023; Guérin et al., 2023; Zhu et al., 2022; Averly & Chao, 2023).

## 3 IMAGENET-OOD: A CLEAN SEMANTIC OOD DATASET

Recent works in OOD detection have primarily focused on failure detection, but new-class detection is still relevant in practice with the growing interest in adaptive learning systems (Kim et al., 2022; He & Zhu, 2022). Consequently, accurate assessment of OOD detection algorithms on semantic shift is very important and needs to be disentangled from covariate shift. Unfortunately, previous datasets were not carefully constructed, leading to contamination from ID classes and unintended covariate shifts. We highlight the shortcomings of past OOD datasets and introduce ImageNet-OOD, a carefully curated, semantic OOD dataset designed to overcome these challenges.

### 3.1 PITFALLS OF EXISTING SEMANTIC SHIFT DATASETS

Early evaluation frameworks in OOD detection primarily use small datasets such as CIFAR-10, CIFAR-100 (Krizhevsky, 2009), SVHN (Netzer et al., 2011), and MNIST (Deng, 2012), but their low-resolution and limited number of classes fail to extend to real-world conditions. Consequently, the outcomes of OOD detection methods in these restricted environments can substantially deviate from those in expansive settings. To include more diverse scenarios, recent OOD detection datasets often designate ImageNet-1K as the ID dataset (Hendrycks et al., 2022; Huang & Li, 2021; Galil et al., 2023). Nevertheless, these contemporary datasets frequently present issues, such as semantic or visual ambiguities and introduction of unnecessary covariate shifts.

**Semantic Ambiguity.** Several existing datasets overlook the hierarchical relations in ImageNet labels, leading to ambiguity in deciding whether a semantic concept is OOD. For example, ImageNet-O (Hendrycks et al., 2021) contains images from the class "pastry dough," which is contained in the "dough" class in ImageNet-1K. We also observe this contamination in OOD datasets that do not utilize ImageNet-21K classes, such as Species (Hendrycks et al., 2022). For example, the ImageNet-1K dataset contains the class "Agaric", which includes the Species class "Agaric Xanthodermus."

**Visual Ambiguity.** Although several datasets remove ID contamination through human filtering, they overlook visual ambiguity attributed to the intricacies in the data collection process of ImageNet. OOD classes can show up in ID images even though the ID and OOD classes are far away on the WordNet semantic tree (Fellbaum, 1998). For example, the C-OOD benchmark (Galil et al., 2023) contains the class "basin," while ImageNet-1K contains the class "lakeside." While "basin" is technically OOD, many images from the "lakeside" and "basin" classes are visually indistinguishable, and thus, can be labeled interchangeably.

**Unnecessary Covariate Shifts.** Several common OOD datasets used when ImageNet-1K is designated as ID include iNaturalist (Horn et al., 2018), SUN (Xiao et al., 2010), and Texture (Cimpoi et al., 2014). However, these datasets come from very specific domains and thus lack the semantic diversity that is reflected in real-world scenarios. Recent works such as NINCO (Bitterwolf et al., 2023) and OpenImage-O (Wang et al., 2022) encourage semantic diversity through manual selection of OOD images. However, due to the limited number of images, limited number of OOD classes, and/or deviation from the original ImageNet data collection procedure, the resulting dataset can potentially introduce hidden covariate shifts that OOD detection algorithms can exploit.

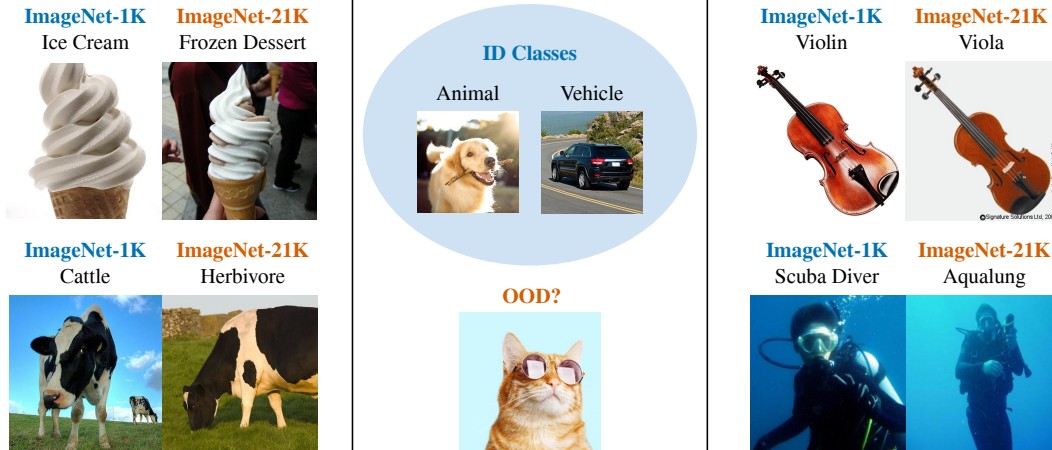

Figure 1: **Removing ambiguities in ImageNet-OOD.** We identify classes in ImageNet-21K which should *not* be included in the ImageNet-OOD dataset, since it would be ambiguous whether they are truly OOD with respect to the ImageNet-1K classes. *Left: Semantic Ambiguity.* "Frozen Dessert" in ImageNet-21K (Hendrycks et al., 2021) should not be considered OOD as it is a hyponym of "Ice Cream." Additionally, classes associated with organism is problematic in the WordNet hierarchy: "Herbivore" contains images from the ImageNet-1K class "Cattle" but it is neither a hypernym or a hyponym. *Middle: Semantically-grounded Covariate Shifts.* A dog vs. vehicle classifier can also be thought of as an animal vs. vehicle classifier. Given this classifier, it is unclear whether "cat" should be considered OOD. *Right: Visual Ambiguity.* "Violin" and "Viola" or "Scuba Diver" and "Aqualung" are visually indistinguishable to human labelers, leading to potential annotation error.

## 3.2 CONSTRUCTION OF IMAGENET-OOD

We start with the 1000 ImageNet-1K classes as ID. These classes are directly sampled from ImageNet-21K, which contains images illustrating the 21K nodes in the WordNet semantic tree (Fellbaum, 1998). We begin the construction of ImageNet-OOD with a pool of candidate classes from the processed version of ImageNet-21K (Ridnik et al., 2021). To reduce semantic ambiguity, we iteratively remove classes based on the following criteria:

**All ImageNet-1K Classes, Their Hypernyms, and Hyponyms.** Classes in ImageNet-1K are simply a subset of classes in ImageNet-21K according to the WordNet semantic tree. Since ImageNet-21K and ImageNet-1K followed the same data collection procedure, the degree of unwanted covariate shift is minimized (Galil et al., 2023). Consequently, these datasets often propose selecting OOD images from the set of ImageNet-21K classes that are disjoint from ImageNet-1K classes (Wang et al., 2018; Hendrycks et al., 2021). However, the hierarchical structure of WordNet allows hypernyms (semantic ancestors) and hyponyms (semantic descendants) of ImageNet-1K classes to contain ID images (*e.g.* "Ice Cream" vs. "Frozen Dessert" in Fig. 1 *Left*). We remove hypernyms and hyponyms of ID classes to promote accurate reflection of OOD detection performance.

**Hyponyms of "Organism".** Natural beings in WordNet and ImageNet-21K are categorized by both technical biological levels and non-technical categories, which leads to inconsistencies in hierarchical relations. For instance, although "Herbivore" is intuitively a hypernym of the ID class "Cattle," such a relationship is not captured by the WordNet semantic tree. Therefore, we avoid this type of ambiguity by removing all the hyponyms of "Organism."

**Semantically-grounded Covariate Shifts.** The definition of "semantic" vs. "covariate" shift becomes ambiguous if the learned decision boundary lies in higher levels of the semantic hierarchy. Consider the scenario in Fig. 1 *Middle*, where "dog" and "vehicle" are the only ID classes. The learned classifier can also be considered "animal" vs. "vehicle" classifier. Then, "cat," although technically a semantic shift, can also be considered a *semantically-grounded covariate shift* with the label "animal." This scenario is very similar to subpopulation shift where bias in the data collection

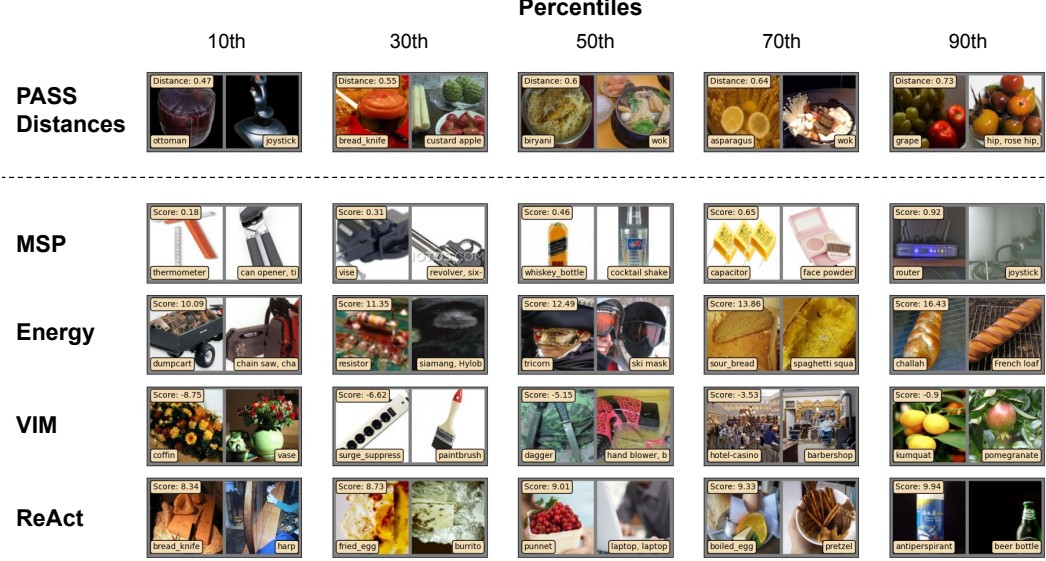

Figure 2: **Examples of Images from ImageNet-OOD.** Images around the 10th, 30th, 50th, 70th, 90th percentile based on either the distance to the closest ImageNet-1K image using features from self-supervised ResNet-50 pre-trained on the PASS dataset (Asano et al., 2021) or scores from OOD detectors MSP (Hendrycks & Gimpel, 2017), Energy (Liu et al., 2020), ViM (Wang et al., 2022), and ReAct (Sun et al., 2021). Within each pair, the left image is the ImageNet-OOD image and the right image is its closest image in ImageNet-1K. These examples illustrate the diversity of ImageNet-OOD and its visual similarity to ImageNet-1K despite having different semantics and OOD scores.

process results in overdomaince of a subclass. Using the WordNet semantic tree, we determine the most general decision boundary for ImageNet-1K. We identify the common ancestor for every pair of ImageNet-1K classes and position each ImageNet-1K class one level below any one of the common ancestors. Subsequently, we redefine all the ImageNet-1K classes to the class defined by this decision boundary during our dataset construction process.

**Final Class Selection.** Despite removing ambiguous classes, manual effort is still needed to resolve visual ambiguities between ImageNet-1K and the rest of ImageNet-21K images. The authors of ImageNet noted that flaws in the construction process of ImageNet-21K can introduce labeling errors due to visual ambiguity across classes (Russakovsky et al., 2015). For instance, the classes "Viola" and "Violin" are virtually indistinguishable unless compared side by side to discern their sizes, thus bringing some degree of human labeling errors. Consequently, the OOD dataset might be contaminated with images from ID classes. Additionally, certain OOD labels are distant on the WordNet semantic tree but contain images with similar semantics. For instance, nearly all images from "Aqualung" (ImageNet-21K) and "Scuba Diver" (ImageNet-1K) depict a scuba diver wearing an Aqualung, since an Aqualung is an apparatus essential for scuba divers to breathe underwater. To avoid the visual ambiguities illustrated in Fig. 1 *Right*, we spent 20 hours to manually select 637 classes from the remaining set that are distinguishable from ImageNet-1K classes by iteratively examining the WordNet neighbors of each ImageNet-1K class and randomly selecting 50 images. Following this, a final 6-hour review is conducted to filter out any images that might have been mislabeled. This process is described in greater details in Section H of the Appendix.

## 3.3 IMAGENET-OOD STATISTICS

ImageNet-OOD contains a total of 31,807 images from 637 classes. Our construction methodology naturally results in a dataset that is as diverse as ImageNet-1K and also very similar visually to ImageNet-1K images. To illustrate this, Fig. 2 displays an amalgamation of images that is very diverse in terms of both semantics and visuals, ranging from fried eggs to capacitors, and objects against plain backgrounds to complex scenes. Additionally, the nearest ImageNet-1K image for each ImageNet-OOD image appears to be in a very similar domain.

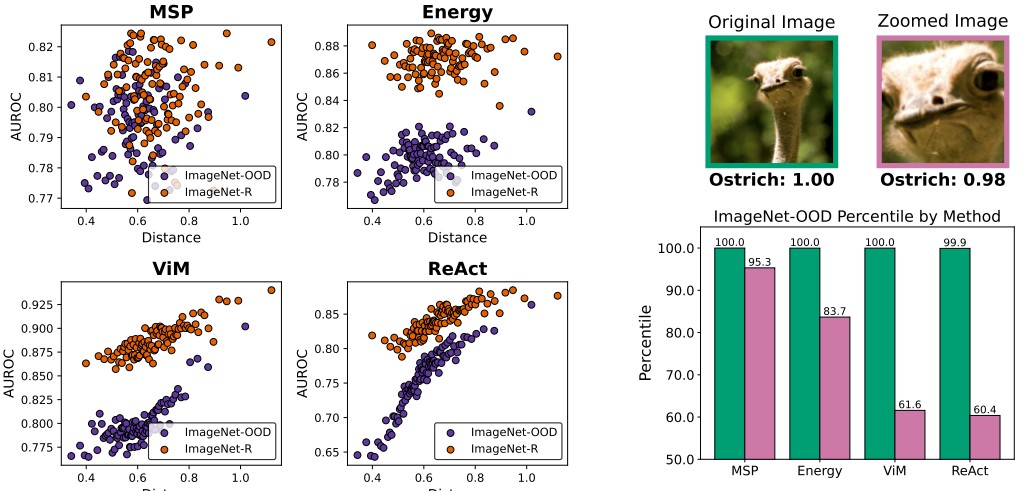

Figure 3: **Influence of Covariate Shift on OOD Detection.** *Left.* Relationship between OOD detection performance and the average distance to the closest ImageNet-1K (Russakovsky et al., 2015) image using features from self-supervised models trained on the PASS (Asano et al., 2021) dataset. Results reveal that given similar PASS feature distances between subsets of the two datasets, modern OOD detection algorithms elicit a stronger response to covariate shift (ImageNet-R (Hendrycks et al., 2020)) than semantic shift (ImageNet-OOD). *Right.* An image of Ostrich in ImageNet-1K dataset where an elementary zoom transformation is applied. The transformation did not influence the model prediction, but substantially decreased the ranking of ViM (Wang et al., 2022) and ReAct (Sun et al., 2021) scores in ImageNet-OOD by 38.4%, 39.6%, respectively.

## 4 EMPIRICAL ANALYSIS

Equipped with ImageNet-OOD, we now analyze the performance of OOD detection algorithms under semantic shift with limited covariate shift. First, we demonstrate that modern OOD detection algorithms are more susceptible to covariate shifts. We reveal that even for images with similar distance to ImageNet-1K, modern OOD detection algorithms perform better at detecting images from covariate shift dataset ImageNet-R (Hendrycks et al., 2020) than images from ImageNet-OOD. Additionally, we design a simple sanity check for OOD detection on random untrained models. Our result show that OOD detection algorithms fails the sanity check and elicit a strong response to covariate shift. Finally, through an extensive evaluation of nine OOD detection algorithms and six datasets over 13 network architectures, we demonstrate that many modern OOD detection algorithms do not draw practical benefits in both new-class detection and failure detection scenarios.

Our experiments include nine logit-based and feature-based OOD detection algorithms, which are more practically adopted due to their require minimal computational cost (Yang et al., 2021b). Logit-based methods MSP (Hendrycks & Gimpel, 2017), Energy (Liu et al., 2020), Max-Cosine (Zhang & Xiang, 2023), and Max-Logit (Hendrycks et al., 2022) derive scoring functions from classification logits because OOD examples tend to have lower activations. Feature-based methods Mahalanobis (Lee et al., 2018), KNN (Sun et al., 2022), ViM (Wang et al., 2022), ASH-B (Djurisic et al., 2022), and ReAct (Sun et al., 2021) operate on the penultimate layer of the model. Hyperparameters are selected based on ablation studies on ImageNet-1K done by original authors.

### 4.1 COVARIATE SHIFTS CONFOUND DETECTION OF SEMANTIC SHIFT

We begin by demonstrating that modern OOD detectors are highly sensitive to covariate shift. Concretely, they cannot *only* detect a large proportion of semantic shift data without simultaneously detecting a large proportion of covariate shift data. To illustrate this, we partition subsets of images with varying degrees of visual similarity to ImageNet-1K, the ID data, with half the subsets exhibiting covariate shift and half exhibiting semantic shift. Specifically, we partition ImageNet-R (Hendrycks et al., 2020) and ImageNet-OOD datasets into 100 subsets each. In each set, images

have similar distances to the closest ImageNet-1K (Russakovsky et al., 2015) image using features from a MoCo-v2 (Chen et al., 2020) self-supervised ResNet-50 (He et al., 2016) model pre-trained on the PASS dataset (Asano et al., 2021), which is an ImageNet replacement derived from YFCC-100M (Thomee et al., 2016). Next, we feed each image into a ResNet-50 classifier trained on ImageNet-1K to obtain the OOD scores. Finally, we calculate the AUROC between images of each subset and images from ImageNet-1K to analyze the relationship between OOD detection performance (AUROC) and the average PASS distance to the closest ImageNet-1K image of each subset.

We make a number of observations shown in Fig. 3 *Left*. First, as one would expect, there is a positive trend between the OOD detection performance and average PASS distance, as images "farther" from the training distribution have been shown to be easier to detect (Fort et al., 2021). The trend seems to be stronger in modern OOD detection algorithms such as Energy (Liu et al., 2020), ViM (Wang et al., 2022), and ReAct (Sun et al., 2021) than in the baseline MSP (Hendrycks & Gimpel, 2017). More importantly, modern detection algorithms are clearly better at detecting ImageNet-R images than ImageNet-OOD images, even with similar PASS distances. To verify this quantitatively, we fit a linear regression model between PASS distance and AUROC for each of the datasets to account for the effect of PASS distance on AUROC. Our results reveal that for ViM, Energy, and ReAct, the 95% confidence intervals on the intercept of the regression trained between the two datasets do not overlap. As a result, there is statistical significance suggesting that the effect of ImageNet-R naturally induces a higher detection performance than ImageNet-OOD on ViM, Energy, and ReAct. The full results and details are included in the Section C of the appendix.

To further demonstrate that OOD scores are affected by covariate shifts, we illustrate that upon introducing a modest covariate shift, there is a notable decline in the scores of OOD detectors. Fig. 3 *Right* displays an image from ImageNet-1K, depicting an ostrich. A ResNet-50 classifier predicts the image correctly with a confidence score around 100%, which ranks higher than 99% of confidence on ImageNet-OOD images. Three modern OOD detectors also assign a score to this image that ranks it higher than 99% of the images from ImageNet-OOD. After applying a zoom to the ostrich, the classifier still exhibits a high classification confidence score of 98%, which still ranks higher than 95.3% of ImageNet-OOD. However, the OOD detection scores rankings from Energy, ViM, and React drops to only 83.7%, 61.6%, and 61.4% higher than the ImageNet-OOD samples, respectively, despite there being no change in semantic label or model prediction.

## 4.2 OOD DETECTION ALGORITHMS FAIL SANITY CHECK ON COVARIATE SHIFT DATASETS

After observing how sensitive modern OOD detection algorithms are to covariate shifts, we design a sanity check to further test this sensitivity in a much stronger setting. Previous works demonstrated the power of random models as feature extractors for tasks such as in-painting, super resolution, and interpretability (Adebayo et al., 2018; Saxe et al., 2011; Alain & Bengio, 2016; Ulyanov et al., 2018). While having this power is acceptable for other tasks, it is very problematic in the context of OOD detection as it challenges the fundamental concept of ID and OOD. The idea of ID becomes ill-defined on random models as the model has not encountered or learned from data sampled from any distributions. Therefore, for a randomly initialized model, the concept of ID should not exist.

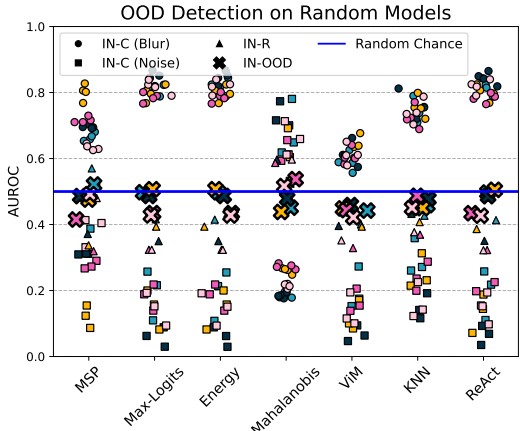

Figure 4: **Performance of OOD detection under random models.** Five ResNet-50 models (indicated by color) with *random* parameters were evaluated on ImageNet-R (IN-R), ImageNet-C (IN-C) and ImageNet-OOD (IN-OOD).

Given that every data point should be considered OOD for a random model, a well-behaved OOD detection algorithm should perform around random chance (AUROC = 0.5) (Hendrycks & Gimpel, 2017). We design a simple sanity check around this idea and found that OOD detection algorithm is biased toward certain covariate shifts. Performance of seven commonly

Table 1: **Performance of OOD Detection Algorithms.** We evaluate seven modern OOD detection algorithms across 13 models, three covariate shift datasets: ImageNet-C (IN-C), ImageNet-R (IN-R), ImageNet-Sketch (IN-Sketch) and two semantic shift datasets: ImageNet-OOD and OpenImage-O, under the goals of both new-class detection and failure detection. Results in each column denotes the AUROC of each pair of **ID** vs. **OOD** datasets. The low AUROCs of the In-C, IN-R, and IN-Sketch vs. IN-OOD experiments indicate that OOD detection algorithms lose their ability to perform new-class detection under the presence of covariate shifts that these datasets introduce. Moreover, the improvement of the best-performing detector is only 0.7% in the IN-1K vs. IN-OOD experiment. The results also confirm that MSP still outperforms all modern methods under failure detection.

| Goal | Method | IN-C IN-OOD | IN-R IN-OOD | IN-Sketch IN-OOD | IN-1K OpenImage-O | IN-1K IN-OOD |
|---|---|---|---|---|---|---|
| New-class Detection | MSP | **55.5** | **48.0** | **46.4** | 84.6 | 79.8 |
| | Max-Logit | 52.3 | 40.4 | 41.0 | 87.9 | 80.5 |
| | Energy | 51.5 | 38.9 | 39.9 | 87.6 | 79.9 |
| | Mahalanobis | 43.6 | 37.8 | 30.7 | 76.0 | 64.8 |
| | ViM | 46.4 | 35.9 | 31.8 | 88.9 | 79.8 |
| | KNN | 39.7 | 36.9 | 26.3 | 76.0 | 59.3 |
| | Max-Cosine | 55.4 | 36.3 | 38.0 | 85.8 | **80.7** |
| | ASH-B | 51.5 | 46.2 | 41.8 | **90.1** | 79.5 |
| | ReAct | 41.4 | 35.0 | 36.9 | 82.7 | 63.4 |
| Failure Detection | MSP | **80.8** | **76.1** | **76.8** | **90.0** | **86.9** |
| | Max-Logit | 76.9 | 68.9 | 72.6 | 89.2 | 84.1 |
| | Energy | 75.5 | 66.4 | 70.9 | 88.5 | 83.1 |
| | Mahalanobis | 57.1 | 58.4 | 51.0 | 74.5 | 65.9 |
| | ViM | 70.7 | 67.1 | 64.7 | 88.7 | 82.5 |
| | KNN | 51.9 | 52.7 | 47.2 | 68.7 | 55.0 |
| | Max-Cosine | 73.5 | 71.5 | 72.6 | 88.5 | 84.8 |
| | ASH-B | 67.7 | 71.7 | 70.4 | 88.3 | 81.0 |
| | ReAct | 60.2 | 47.9 | 55.9 | 80.4 | 65.7 |

used OOD detection methods is evaluated using five random models on the ImageNet-R (Hendrycks et al., 2020) images and on blurring and noise corruptions in ImageNet-C (Hendrycks & Dietterich, 2019). This experiment is performed on Resnet-50 with Kaiming normal initialization (He et al., 2015). We use only the most severe corruptions in ImageNet-C.

Fig. 4 reveals that OOD detectors confidently detect blurry ImageNet-C images as OOD (AUROC > 0.5) and noisy ImageNet-C images as ID (AUROC < 0.5). Additionally, they also tend to detect ImageNet-R images as ID. ImageNet-OOD, on the other hand, is unaffected and detected around random chance (AUROC = 0.5). The bias toward detecting certain corrupted images illustrates that OOD detectors can easily pick up patterns from covariate shift, even on untrained models.

## 4.3 MODERN OOD DETECTION ALGORITHMS DO NOT BRING PRACTICAL BENEFITS

In this section, we show that modern detection algorithms do not gain significant benefits over the MSP (Hendrycks & Gimpel, 2017) baseline regardless of the approach to OOD detection. Under new-class detection, we reveal that when covariate shifts are minimized, modern detection algorithms have less than 1% AUROC improvement over the MSP baseline. Furthermore, comprehensive analysis under failure detection reveals that modern OOD detection algorithms do not actually improve at distinguishing between correctly classified examples and semantically OOD examples.

For the evaluation, we used 13 different convolutional neural networks trained on ImageNet-1K from the torchvision library: ResNet(He et al., 2016), DenseNet (Huang et al., 2017), Wide ResNet (Zagoruyko & Komodakis, 2016), RegNet (Xu et al., 2022), ResNeXt (Xie et al., 2017). The algorithms were evaluated across five different datasets: ImageNet-C (Hendrycks & Dietterich, 2019), ImageNet-R (Hendrycks et al., 2020), ImageNet-Sketch(Wang et al., 2019), OpenImage-O (Wang et al., 2022), and ImageNet-OOD. The results from Table 1 reports the average AUROC across the 13 models under both new-class and failure detection scenarios.

**New-class Detection.** New-class detection aims to detect only examples from semantic shift datasets. All images that belong to training classes, including covariate shifted examples, are considered ID. We first demonstrate that under the influence of certain covariate shifts, OOD detection algorithms are not able to identify semantic shifts. Table 1 displays AUROC scores on three covariate shift datasets vs. ImageNet-OOD. All detection algorithms show AUROC below 50% on ImageNet-Sketch and ImageNet-R vs. ImageNet-OOD, indicating a less than 50% probability that an OOD detector scores an ImageNet-OOD example higher than an ImageNet-Sketch example. However, MSP yields significantly higher AUROC than other OOD detectors, revealing that modern OOD detectors are more susceptible towards covariate shift.

Next, we demonstrate that improvements gained on past datasets disappear when modern OOD detection algorithms are evaluated on ImageNet-OOD. The discrepancy in AUROC between ImageNet-1K vs. OpenImage-O and ImageNet-1K vs. ImageNet-OOD highlights the importance of the semantic shift dataset. In particular, on OpenImage-O, Max-Logit, ViM, and ASH-B all exhibit significant improvements over the MSP baseline, showing a 3.3%, 4.3%, and 5.5% enhancement in AUROC, respectively. However, when evaluated on ImageNet-OOD, the improvements drastically shrank to 0.7% for Max-Logit, completely disappears for ViM and even decreased by 0.3% for ASH-B. We provide qualitative explanations for this phenomenon in Section G in the Appendix. With the insignificant improvement under semantic shift coupled with the increased susceptibility towards covariate shift, it is unclear whether modern OOD detection algorithms yields any practical improvements over the baseline when deployed to the real-world.

**Findings on Failure Detection.** Failure detection aims to detect incorrectly classified examples regardless of the type of distribution shift with semantic shift examples being incorrect by definition. Results from Table 1 confirms previous findings that MSP outperforms modern OOD detection algorithms under failure detection across all experiments. Interestingly, on the ImageNet-1K vs. ImageNet-OOD experiment, MSP outperforms Max-Logit by 2.8% under failure detection despite Max-Logit outperforming MSP by 0.7% under new-class detection. This discrepancy is particularly an unexpected outcome since semantic shift examples are considered OOD for both failure and new-class detection. Breaking down the test data by whether or not they are correctly predicted resolves this mystery. Fundamentally, the goal of an OOD detection algorithm is to

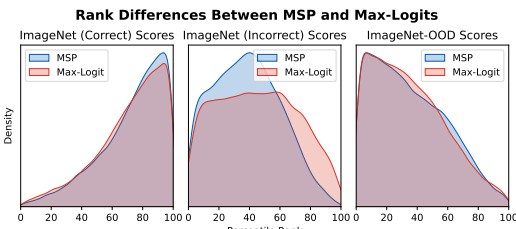

Figure 5: **Comparison of the ranking between MSP and Max-Logit .** *Left.* MSP is slightly better at ranking correctly predicted ImageNet images higher. *Center.* Max-Logit ranks more incorrect ImageNet images higher than MSP. *Right.* MSP and Max-Logits have near identical ranks on ImageNet-OOD examples.

give lower scores, and hence, lower rankings, on semantic shift data (Equation 1). However, we observe from Fig. 5 that Max-Logit does not rank the semantic shift (ImageNet-OOD) examples lower (more OOD) than MSP does. Instead, it ranks the incorrect test (ImageNet-1K) examples higher. In other words, Max-Logit is neither better at detecting semantic shift examples (Fig. 5 *Right*) nor better at preserving correct test examples (Fig. 5 *Left*). Instead, it is better at preserving incorrect test examples (Fig. 5 *Center*), which is the *opposite* of the ideal behavior for catching model failures.

## 5 CONCLUSION

We introduce ImageNet-OOD, a carefully curated, diverse OOD detection dataset for studying the effects of semantic shifts. By building on top of ImageNet-21K and manually selecting test classes to remove semantic and visual ambiguities, we avoid unnecessary covariate shifts that may confound the performance of OOD detection algorithms. Using ImageNet-OOD, we reveal that many modern OOD detection algorithms detect covariate shifts to a greater extent than semantic shifts and further demonstrate that the improvement of many algorithms disappears. We hope that our dataset and findings will help the OOD detection community in building methods that are more effective at detecting semantic shifts and aligning their behavior with their stated purpose.

## ETHICS STATEMENT

Although our dataset and analysis does not directly produce harmful impact, we note that the bias from ImageNet may be propagated as ImageNet-OOD reuses images from ImageNet-21K (Deng et al., 2009; Yang et al., 2020). As a means of mitigation, we manually filter out inappropriate classes. In addition, OOD detection becomes a high-stake task to prevent catastrophic failures in computer vision systems. While we strive to include a diverse set of classes in ImageNet-OOD, we cannot guarantee that performance on ImageNet-OOD is the a valid indicator for all safety-critical applications.

## REPRODUCIBILITY STATEMENT

We provide the class names and synset IDs for all ImageNet-OOD classes in the appendix. In addition, we provide all image filenames as presented in the ILSVRC 2012 Challenge (Russakovsky et al., 2015), as well as sample code for the experiments in the linked Github repository linked in the abstract.

## ACKNOWLEDGEMENTS

This material is based upon work supported by the National Science Foundation under Grant No. 2112562. Any opinions, findings, and conclusions or recommendations expressed in this material are those of the authors' and do not necessarily reflect the views of the National Science Foundation. We also thank the Princeton VisualAI Lab members and Christiane Fellbaum for helpful feedback.

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

APPENDIX

In this appendix, we will provide more detailed analysis on claims made in the main paper.

- **Section A**: We provide an related work on covariate shifts in OOD detection, failure detection, and different OOD detection datasets used in the paper.
- **Section B**: We extend Section 4.1 of the main paper to analyze the difference between score distributions across multiple OOD detection algorithms.
- **Section C**: We supplement Section 4.1 of the main paper on quantitative results using PASS (Asano et al., 2021) distances.
- **Section D**: We analyze the behavior of methods developed for Open-Set Recognition and gradient-based OOD detection.
- **Section E**: We supplement Section 4.3 of the main paper and provide more analysis on the separability between correct and incorrect ID examples.
- **Section F**: We supplement Section 4.2 of the main paper and perform the sanity check to examine covariate shift bias from different model architectures.
- **Section G**: We supplement Section 4.3 of the main paper to provide qualitative examples of images revealing the inherent bias of modern detectors towards certain covariate shift.
- **Section H**: We supplement Section 3 of the main paper and provide details on the construction process of the ImageNet-OOD dataset.
- **Section I**: We supplement Section 3 of the main paper and provide class names and synset IDs for the 637 ImageNet-OOD classes.

## A    RELATED WORK

**Covariate shift in OOD detection.**    Covariate shift was first considered in OOD detection by Generalized ODIN (Hsu et al., 2020), where OOD detection performance was evaluated considering all covariate shifted examples as out-of-distribution. (Tian et al., 2021) designed a scoring function that disentangles detection of semantic vs. covariate shifts. Later, (Yang et al., 2023) pointed out that models should ideally generalize, instead of detect, in the case of covariate shifts, because generalization is the primary goal of machine learning. Thus, they defined all covariate shifted examples as in-distribution and proposed several benchmarks that include covariate shifted data. Recently proposed benchmark OpenOODv1.5 (Zhang et al., 2023) coined the term full-spectrum detection to encapsulate this idea. They found that under this setting, OOD detection performance are significantly hindered in contrast to the traditional setting. While new-class detection characterizes all covariate shifted examples as ID, failure detection only characterize the correctly classified ones as ID, since the rejection of covariate shift examples that would otherwise hurt the model's classification performance should not be penalized.

**Failure Detection.**    OOD detection is often categorized as a sub-task of failure detection, as the primary goal of OOD detection is to catch unsafe prediction before models make a mistake. Other tasks that share this common goal of failure detection include misclassification detection (Hendrycks & Gimpel, 2017) and uncertainty calibration (Kendall & Gal, 2017), both of which differs from OOD detection in that they do not consider examples outside the set of training classes. However, since all failure detection methods can be interpreted as scoring functions and are motivated from a common principle (alerting failure before occurrence), (Jaeger et al., 2023) argues that all failure detection tasks should be evaluated across a common benchmark that takes into account the classification performance of the model. (Xia & Bouganis, 2022) made similar arguments under Selective Classification in the presence of OOD Data (SCOD).

**Datasets for OOD detection.**    In the experiments of the paper, we used three covariate shift datasets: ImageNet-R (Hendrycks et al., 2020), ImageNet-C (Hendrycks & Dietterich, 2019), and ImageNet-Sketch (Wang et al., 2019). ImageNet-R has 30,000 images containing renditions of 200 ImageNet-1K classes. These renditions include art, cartoons, deviantart, graffiti, embroidery, graphics, origami, paintings, patterns, plastic objects, plush objects, sculptures, sketches, tattoos,

Distribution of OOD Scores for ImageNet-C

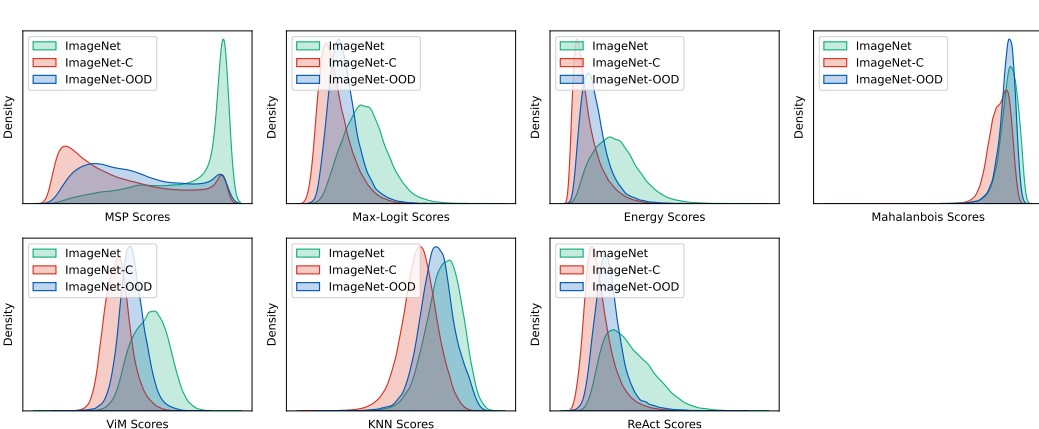

Figure 6: **Distribution of OOD scores on ImageNet-C.** A kernel density estimator further illustrates that all OOD detectors detect covariate shift. Results reveal that for both covariate shift datasets ImageNet-C, the distribution of scores is lower than that of semantic shift.

Distribution of OOD Scores for ImageNet-R

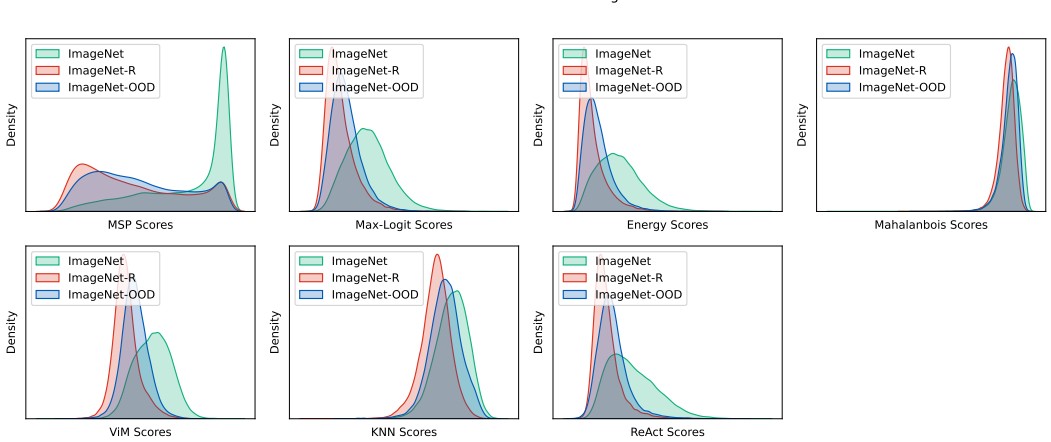

Figure 7: **Distribution of OOD scores on ImageNet-R.** Same setup as Figure 6 but on ImageNet-R and reaches the same conclusion.

toys, and video games. ImageNet-C includes corrupted versions of ImageNet-1K images at different severity levels using corruptions such as brightness, contrast, elastic, pixelate, and JPEG. ImageNet-Sketch contains 50,000 images of sketches of all ImageNet-1K classes. We also used OpenImage-O (Wang et al., 2022), a manually curated semantic shift dataset, which is derived from OpenImages (Kuznetsova et al., 2020), an object detection dataset.

## B  ANALYSIS OF SCORE DISTRIBUTIONS

In the following experiment, we will compare the score distributions of OOD detectors on semantic and covariate shift datasets to further show that that OOD detectors cannot *only* reject a large proportion of semantic shift data without simultaneously rejecting a large proportion correctly classified covariate shift data.

We provide complete analysis across seven OOD detection algorithms on their distribution of scores between covariate and semantic shift. Fig. 6 and 7 shows that all seven algorithms rank a significant proportion of covariate shift examples lower than semantic shift examples. Additionally, breaking down covariate shift examples between correct vs. incorrect, Fig. 8 and 9 reveals that all seven

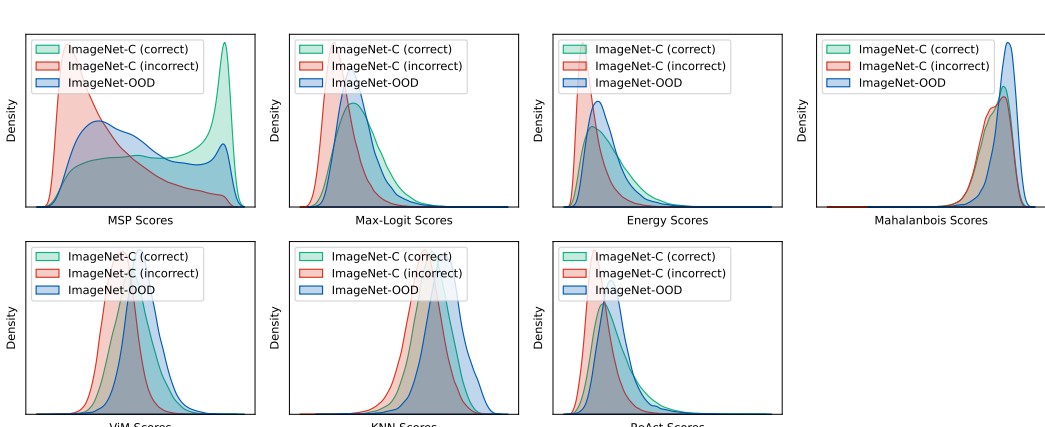

Figure 8: **Score breakdown on covariate shift data on ImageNet-C.** Comparison of detection score distributions for correctly classified covariate shift examples on ImageNet-C and semantic shift examples (ImageNet-OOD). Scores of correct ImageNet-C examples tend to be even lower than ImageNet-OOD examples. Rejecting a significant portion of semantic shift data leads to the rejection of a significant portion of correct covariate shift data.

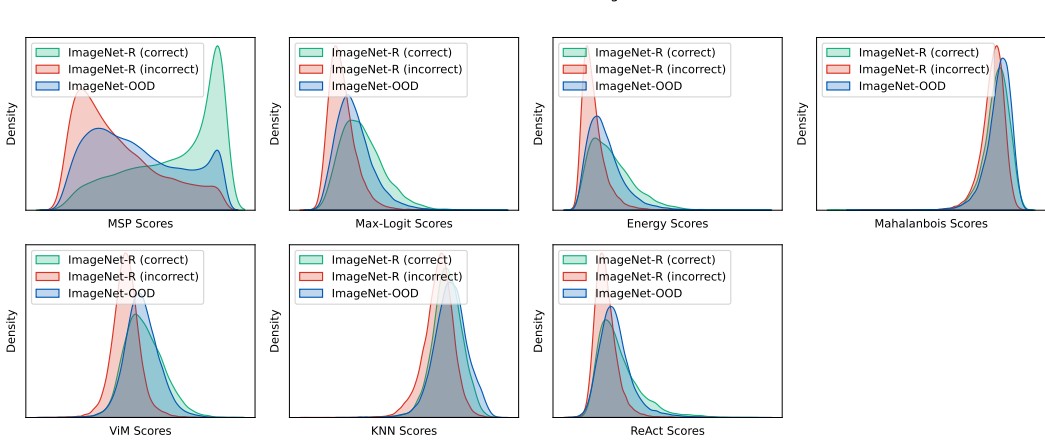

Figure 9: **Score breakdown on covariate shift data on ImageNet-R.** Same setup as Figure 8 but on ImageNet-R. Scores of correct ImageNet-C examples tend to be similar to ImageNet-OOD examples.

Table 2: **Proportion of correct examples discarded by rejecting 75 percent of ImageNet-OOD examples.** Using the threshold where 75 percent of ImageNet-OOD examples are rejected as OOD, how many correct examples from ImageNet-1K, ImageNet-C, and ImageNet-R are discarded, which hurt model safety. Results reveals that modern OOD detection methods does not show significant lower false rejection rate on ImageNet-1K and have significantly worse false rejection rate when covariate shift is introduced by ImageNet-C and ImageNet-R.

| | ImageNet (Russakovsky et al., 2015) | ImageNet-C (Hendrycks & Dietterich, 2019) | ImageNet-R (Hendrycks et al., 2020) |
|---|---|---|---|
| MSP (Hendrycks & Gimpel, 2017) | 18 | **53.6** | **41.9** |
| Max-Logit (Hendrycks et al., 2022) | **17.8** | 65.6 | 58.7 |
| Energy (Liu et al., 2020) | 19.3 | 67.4 | 61.2 |
| Mahalanobis (Lee et al., 2018) | 59.0 | 91.7 | 80.2 |
| ViM (Wang et al., 2022) | 22.2 | 85.3 | 68.8 |
| KNN (Sun et al., 2022) | 57.8 | 92.3 | 83.1 |
| ReAct (Sun et al., 2021) | 27.1 | 73.5 | 68.0 |

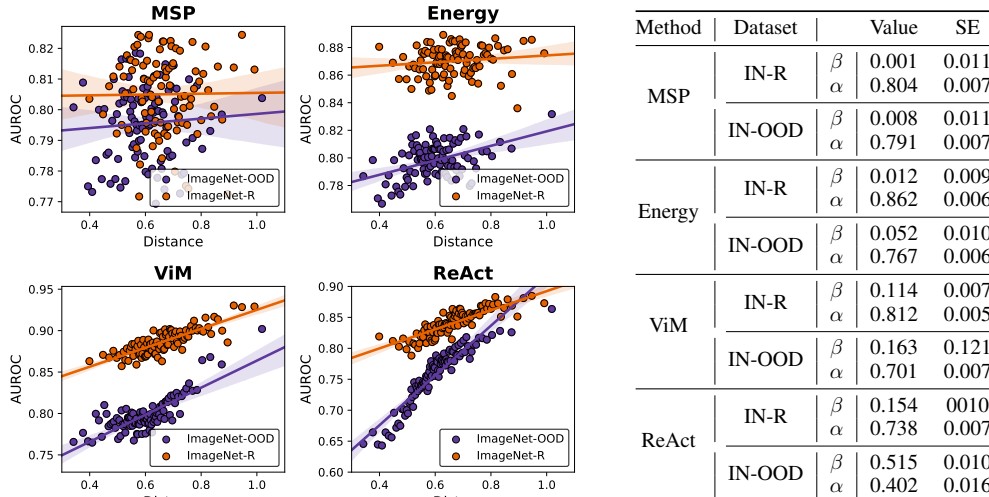

| Method | Dataset | | Value | SE |
|--------|---------|-----|-------|-----|
| MSP | IN-R | $\beta$ | 0.001 | 0.011 |
| | | $\alpha$ | 0.804 | 0.007 |
| | IN-OOD | $\beta$ | 0.008 | 0.011 |
| | | $\alpha$ | 0.791 | 0.007 |
| Energy | IN-R | $\beta$ | 0.012 | 0.009 |
| | | $\alpha$ | 0.862 | 0.006 |
| | IN-OOD | $\beta$ | 0.052 | 0.010 |
| | | $\alpha$ | 0.767 | 0.006 |
| ViM | IN-R | $\beta$ | 0.114 | 0.007 |
| | | $\alpha$ | 0.812 | 0.005 |
| | IN-OOD | $\beta$ | 0.163 | 0.121 |
| | | $\alpha$ | 0.701 | 0.007 |
| ReAct | IN-R | $\beta$ | 0.154 | 0010 |
| | | $\alpha$ | 0.738 | 0.007 |
| | IN-OOD | $\beta$ | 0.515 | 0.010 |
| | | $\alpha$ | 0.402 | 0.016 |

Figure 10: **Full Analysis on influence of Covariate Shift on OOD Detection.** *Left.* Relationship between OOD detection performance and the average distance to the closest ImageNet-1K (Russakovsky et al., 2015) image using features from self-supervised pre-trained models on the PASS (Asano et al., 2021) dataset. This figure adds a linear regression model with its 95% confidence interval to Fig. 3. The results augments findings in section 4.1 by revealing substantial overlap in confident region in MSP but low overlap for Energy, ViM, and ReAct. *Right.* Quantitative measures for each of the fitted linear model with given form AUROC $= \beta$(distance) $+ \alpha$. The 95% confidence interval for the $\alpha$ between the two datasets do not overlap for Energy, ViM, and ReAct, but do overlap for MSP. This means that the difference in intercept coefficient is statistically significant for Energy, ViM, and ReAct but not MSP.

algorithms do indeed confuse correctly classified covariate shift examples with semantic shift examples. Finally, Table 2 uncovers that all seven algorithms yield significantly more rejection of correctly classified examples with covariate shift datasets. This provide comprehensive support for the conclusion drawn in Section 3.2 of the main paper.

## C  QUANTITATIVE ANALYSIS ON PASS DISTANCES

In this section, we provide full details on the quantitative measures of the relationship between PASS (Asano et al., 2021) distance (image similarity) and AUROC (OOD detection performance). The results in Fig. 10 *Right* confirms that there is a statistical significant difference on the performance of OOD detection between covariate shift (ImageNet-R) and semantic shift (ImageNet-OOD). Using the standard error, we can derive the 95% confidence interval for the intercept measure of the linear regression model. For example, using two times the standard error, we can derive for energy the respective confidence interval between ImageNet-R and ImageNet-OOD as (0.85, 0.874) and (0.755, 779). Since these two intervals do not overlap, there is statistical significance that the real intercept between the two datasets is different. Intuitively, the intercept of the model measure the effect of dataset independent from PASS distance on OOD detection performance. Since it appears the intercept for the ImageNet-R model is higher, this suggests covariate shifts tend to boost AUROC, demonstrating that OOD detectors detect covariate shift more easily. Similar conclusions can be reached for ViM and ReAct, but not MSP, revealing modern algorithms are more susceptible to covariate shifts. Fig. 10 *Left* unravel the performance different between the two datasets across different PASS measures. We observe that the confident region of the linear regression model have substantial overlaps for MSP and minor overlap for ReAct. On other hand, Energy and ViM have no overlap within the specified interval, indicating that the modern detectors are more affected by covariate shift than MSP.

Table 3: **OOD detection performance for ResNet-50 under Gradient-based OOD and Open-Max.** OOD detection performance for Open-Set Recognition and gradient-based OOD detection methods on the two semantic shift datasets suggest the same conclusion: MSP baseline is best for failure detection and modern methods do not have substantial improvement over baseline once spurious covariate shift is removed. ODIN$_\epsilon$ uses $T = 1000$ and $\epsilon = 0.0014$. ODIN uses $T = 1000$ with no adversarial perturbations. GradNorm uses $T = 1$.

| Goal | Method | OpenImage-O | ImageNet-OOD |
|------|--------|-------------|--------------|
| New-class Detection | MSP | 84.0 | 79.2 |
| | ODIN$_\epsilon$ | 86.5 | 80.1 |
| | ODIN | 87.4 | **80.4** |
| | GradNorm | 80.4 | 73.8 |
| | OpenMax | **87.4** | 80.2 |
| Failure Detection | MSP | **89.9** | **86.8** |
| | ODIN$_\epsilon$ | 87.9 | 83.5 |
| | ODIN | 89.3 | 84.6 |
| | GradNorm | 77.2 | 72.2 |
| | OpenMax | 88.5 | 83.4 |

Table 4: **Detailed breakdown of AUROC performance with respect to ImageNet-OOD.** Breakdown reveals that the improvement observed in both Max-Logit and Energy is attributed to better separation between incorrect ID predictions and OOD predictions, shown by the large margin of increase in AUROC between Incorrect ID (+) vs. OOD (-). Performance between correct in-distribution predictions and out-of-distribution predictions are similar. Additionally, when evaluating incorrect ID predictions vs. correct ID predictions, separability decreases for Max-Logit and Energy compared to MSP.

| | Correct ID (+) vs. OOD (-) | Incorrect ID (+) vs. OOD (-) | Correct ID (+) vs. Incorrect ID (-) |
|---|---|---|---|
| MSP (Hendrycks & Gimpel, 2017) | 86.9 | 54.7 | 86.4 |
| Max-Logit (Hendrycks et al., 2022) | 86.3 | 61.8 | 77.7 |
| Energy (Liu et al., 2020) | 85.4 | 62.4 | 76.2 |

# D  OPEN-SET RECOGNITION AND GRADIENT-BASED OOD DETECTION

Open-Set Recognition (OSR) is a similar task to semantic OOD detection, where the goal is to identify unseen classes. Gradient-based OOD detection algorithms uses some aspects on the gradients of a model to calculate an OOD score. Since detection requires the calculation of gradient on each image, such method of OOD detection is more computationally expensive. We perform the same analysis from 4.3 but on a single ResNet-50 model on the popular OSR algorithm Open-Max (Hsu et al., 2020) which does not require any retraining and gradient-based OOD detection algorithm ODIN (Liang et al., 2018) and GradNorm (Huang et al., 2021) on OpenImage-O and ImageNet-OOD. Results from Table 1 reveals the same consistent behavior where we see substantial improvement from MSP on OpenImage-O but not ImageNet-OOD for new-class detection, and the MSP baseline outperforms these methods for failure detection.

# E  ADDITIONAL EXPERIMENTS FOR DETECTION OF INCORRECT EXAMPLES

In addition to the analysis that visualizes the ranking of OOD scores in Section 4.3, we quantitatively examine AUROC on ImageNet-OOD to measure the separability between three groups of examples: correctly classified in-distribution (ID) examples, incorrectly classified ID examples, and semantic shifted OOD examples.

Specifically, since AUROC is a measure of separability (*i.e.* the probability of scoring a positive example higher than a negative example), we examine the AUROC among these three groups of examples. Using the probabilistic interpretation of AUROC, we can decompose AUROC between

ID vs. OOD by correct ID and incorrect ID using the law of total probability:

$$p(f(x_{in}) > f(x_{out})) =$$
$$p(f(x_{in}) > f(x_{out})|C(x_{in}) = 1)p(C(x_{in}) = 1) + \qquad (2)$$
$$p(f(x_{in}) > f(x_{out})|C(x_{in}) = 0)p(C(x_{in}) = 0)$$

where $x_{in}$, $x_{out}$ refer to semantic ID and OOD examples, respectively, $f$ is the OOD scoring function, and $C$ is an indicator function with $C(x) = 1$ if x is predicted correctly, and 0 otherwise.

Table 4 reports $p(f(x_{in}) > f(x_{out})|C(x_{in}) = 1)$ (*i.e.* column Correct ID (+) vs. OOD (-)) and $p(f(x_{in}) > f(x_{out})|C(x_{in}) = 0)$ (*i.e.* column Incorrect ID (+) vs. OOD (-)). The results reveal increased separability between incorrect ID vs. OOD from MSP to ViM and ReAct, decreased separability between correctly classified ID vs. incorrectly classified ID, and roughly the same separability between correctly classified vs. OOD. Since $p(C(x_{in}) = 1)$ in Equation 2 is the classification accuracy on the ID dataset, we see that for $71.6\%$ of increase in new-class detection AUROC from ViM can be attributed to an increase in performance of detecting Incorrect ID vs. OOD predictions. This pattern supports the claim that many advanced OOD detection methods improve under the old benchmark by detecting more incorrectly classified examples as ID rather than a balanced improvement across the ID set.

Additionally, we also use AUROC to approximate the probability of scoring correct ID higher than incorrect ID examples, reported in the Correct ID (+) vs. Incorrect ID (-) column in Table 4. We found that advanced OOD detection methods have lower separability between correctly classified ID vs. incorrectly classified ID. Having lower separability between correct vs. incorrect hurts the performance of the model from a safety perspective, as more problematic examples can pass through the OOD detection filter.

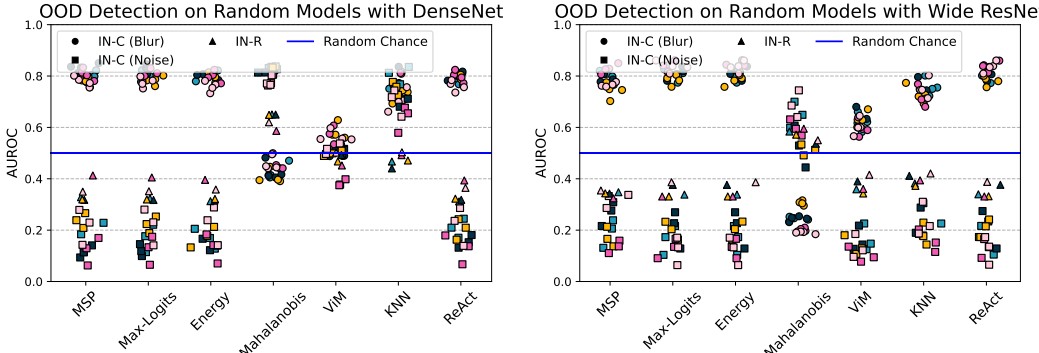

Figure 11: **OOD detection performance under random models.** AUROC performance of 5 DenseNet-121 models (*left*) or 5 Wide ResNet-50 model (*right*) with **random**, untrained parameters on subsets of ImageNet-C (Hendrycks & Dietterich, 2019) under the existing benchmark vs our proposed benchmark. Colors indicate the specific random model and the markers indicate the corruption type. Results reveals the same conclusion as the ResNet-50 sanity check.

## F  SANITY CHECK EXTENDS TO OTHER MODEL ARCHITECTURES

We expand our analysis of the sanity check in Section 4.2 to other model architectures and to the scenario of failure detection. For architectures, we display additional random DenseNet-121 (Huang et al., 2017) and Wide ResNet-50 (Zagoruyko & Komodakis, 2016). Results from Figure 11 reveal that the same issue with sanity check occurs on DenseNet, except ViM, and Wide Resnet-50 in the scenario of new-class detection, suggesting that this issue applies to convolution based architecture.

## G  QUALITATIVE ANALYSIS ON COVARIATE SHIFT BIAS

We expand our analysis by visualizing images where modern OOD detection algorithms and the baseline MSP differ the most. We perform this analysis on ImageNet-1K, ImageNet-OOD, OpenImage-O, and ImageNet-Sketch on a ResNet-50 model trained on ImageNet-1K. We found that modern OOD detection algorithms ViM and ASH-B tends to latch on to specific spurious features that are exacerbated by dataset with uncontrolled covariate shift, confounding their evaluation on semantic shift.

We reveal in Figure 12 that the OOD detection method ViM tends to give texture-like images low scores, detecting them as OOD. This effect is more prominent in OpenImage-O because it appears there are many images in a different domain. In contrast, our dataset ImageNet-OOD has images more similar to ImageNet-1K. The observed change in domain is a clear example of unnecessary covariate shifts, explaining the vanishing performance gains of ViM in ImageNet-OOD. Similarly in Figure 13, we observe that the OOD detection algorithm ASH-B tends to score text-like images as OOD. Though there are some imperfections due to the class semantics (e.g. map is an object but also can be a diagram), ImageNet-OOD overall still produced more similar looking images to ImageNet-1K.

In summary, qualitative analysis on the discrepancy between modern OOD detection algorithms and MSP motivate our dataset: the minimization of covariate shift is important when assessing an OOD detector's performance on semantic shift for the task of new-class detection.

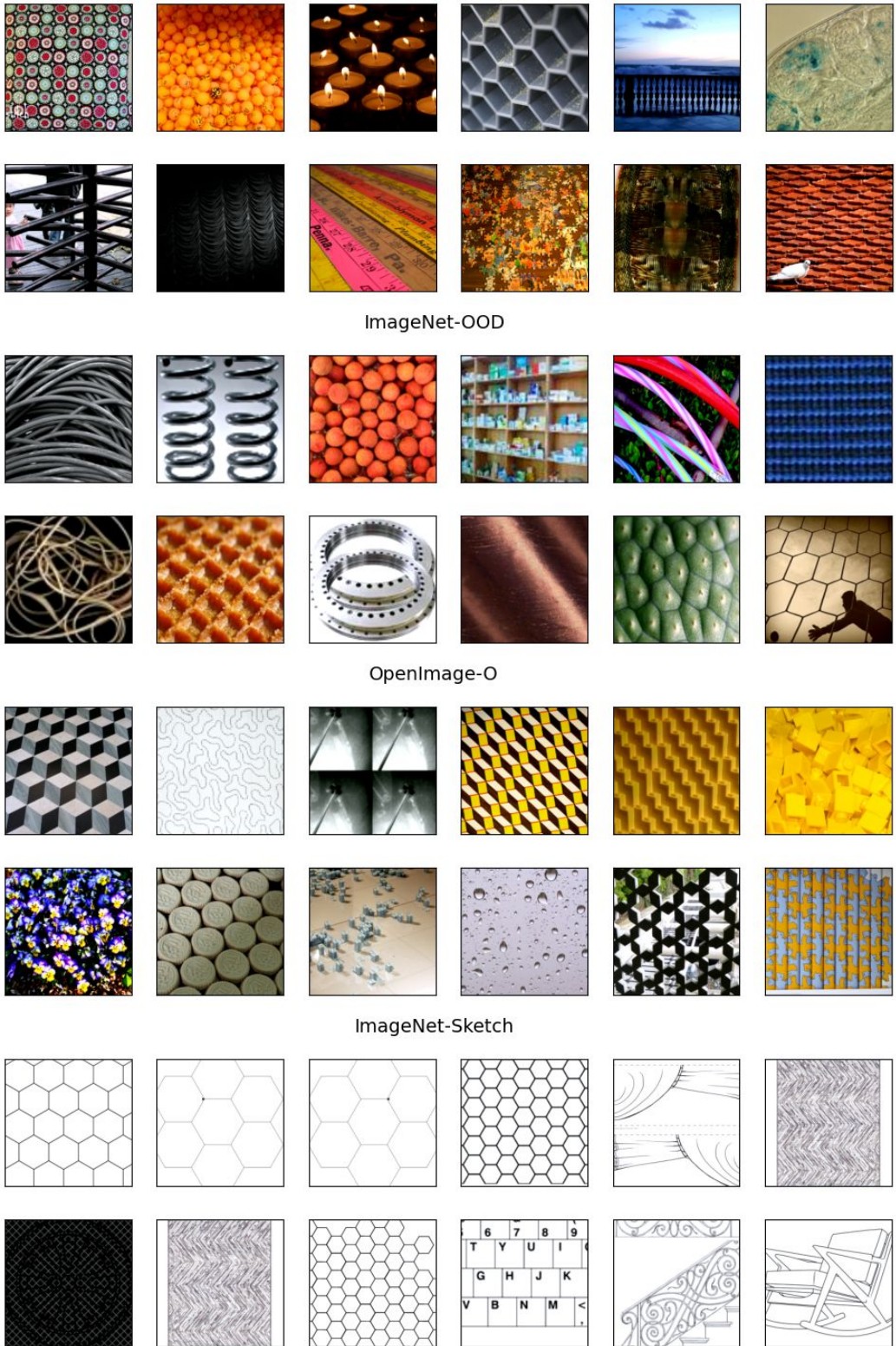

Figure 12: **Images with highest discrepancy in ranks between ViM and MSP.** We show 12 images from ImageNet-1K, ImageNet-OOD, OpenImage-O, and ImageNet-Sketch with the highest discrepancy in ranks where ViM scores low (OOD) and MSP scores high (ID) revealing that ViM tends to prefer detecting texture as OOD. The images also reveal that ImageNet-OOD images are more realistic and comparable to ImageNet-1K than OpenImage-O.

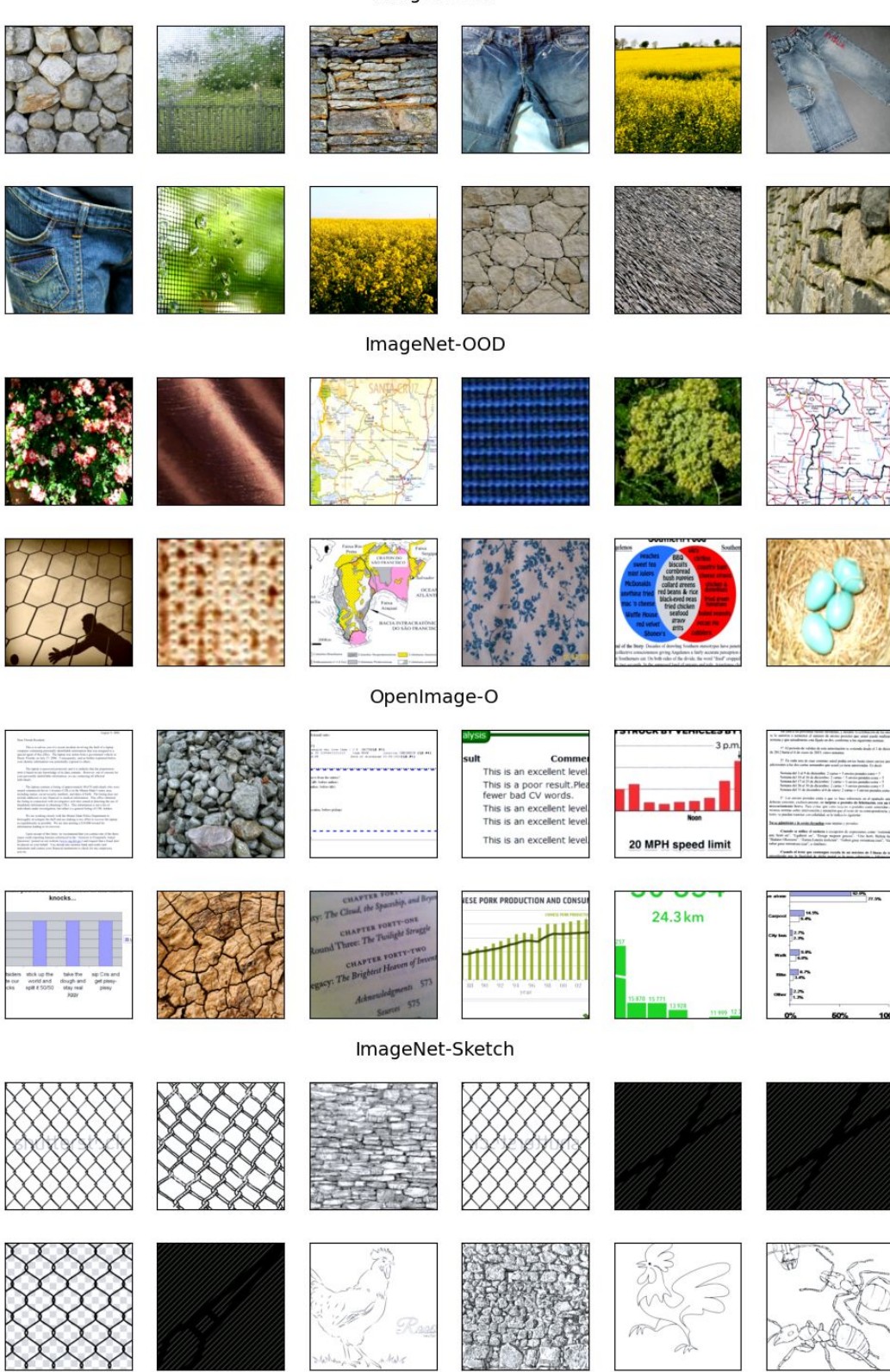

Figure 13: **Images with highest discrepancy in ranks between ASH-B and MSP.** We show 12 images from ImageNet-1K, ImageNet-OOD, OpenImage-O, and ImageNet-Sketch with the highest discrepancy in ranks where ASH-B scores low (OOD) and MSP scores high (ID) revealing that ASH-B tends to prefer detecting text as OOD.

## H  MORE DETAILS ON IMAGENET-OOD CONSTRUCTION

In this section, we provide details on the manual selection process of ImageNet-OOD. Because images from ID classes may leak into OOD classes if human labelers are unable to disambiguate two classes, such as "violin" and "viola" (Russakovsky et al., 2015), we manually selected 1000 classes from ImageNet-21K to construct ImageNet-OOD. However, even after excluding hypernyms, hyponyms, and the "organism" subtree, there are still 5074 remaining candidate ImageNet-21K classes. It is simply infeasible to check all 5074 classes against all 1000 ImageNet-1K classes, which would require $5074 \times 1000 = 5,074,000$ manual comparisons. Therefore, we need to employ another mechanism that can pass through the 5074 classes in linear time.

To start the collection process, we aim to pick out 1000 classes, We first gathered the sister classes for each ImageNet-1K class. A sister class $c_s^i$ is defined as a class that shares a direct parent with an ImageNet-1K class $c^i$. For example, the sister classes for the ImageNet-1K class "microwave" has sister classes "food_processor", "ice_maker", "hot_plate", "coffee_maker", and "oven", because these classes all have the same direct parent "kitchen_appliance." Considering only sister classes allowed us to further reduce the search space down to 2874 candidate classes.

Once we had obtained the sister classes, we examined the visual and semantic ambiguity between each sister class and its corresponding ImageNet-1K class through example images. Unambiguous classes are added to the final list of ImageNet-OOD classes. Since the ambiguity between classes was considered during the curation of ImageNet-1K classes, we assume that there exists minimal ambiguity between classes under different subtrees that contains an ImageNet-1K class. This assumption allowed us to only examine the relationship of sister classes with their corresponding ImageNet-1K class instead of with all 1000 ImageNet-1K classes. In the end, we only needed to compare 13,831 pairs of classes.

Following the manual selection of the 1000 classes, we filtered out the classes with semantically-grounded covariate shift using the method described in Section 3. Then, we examined the 1000 images that are the closest to ImageNet-1K validation images in terms of ResNet-50 (He et al., 2016) feature distance to filter out visual similarity. We filter out both classes and images. Classes such as "aqualung" (visually similar to ID class "scuba diver") were filtered out in this stage, and images with indistinguishable visuals were also thrown out. The final resulting dataset will include 31,807 images from 637 classes.

## I  IMAGENET-OOD CLASSES

| Synset ID | Class Name | Synset ID | Class Name |
|-----------|------------|-----------|------------|
| n02666943 | abattoir | n04108822 | rope_bridge |
| n02678897 | adapter | n04113406 | roulette_wheel |
| n02688273 | air_filter | n04114844 | router |
| n02689434 | air_hammer | n04116098 | rubber_band |
| n02698634 | alpenstock | n04118635 | ruin |
| n02705429 | amphora | n04119230 | rumble_seat |
| n02705944 | amplifier | n04122685 | sachet |
| n02710044 | andiron | n04123740 | saddle |
| n02723165 | antiperspirant | n04132603 | samisen |
| n02725872 | anvil | n04134008 | sandbag |
| n02726681 | apartment_building | n04136800 | sash_fastener |
| n02757337 | audiometer | n04139140 | saucepot |
| n02758960 | autoclave | n04150153 | scouring_pad |
| n02763604 | aviary | n04150980 | scraper |
| n02767956 | backbench | n04167346 | seeder |
| n02768226 | backboard | n04168199 | Segway |
| n02770721 | backscratcher | n04171831 | semiconductor_device |
| n02770830 | backseat | n04176068 | serving_cart |
| n02775897 | Bailey_bridge | n04176190 | serving_dish |
| n02776205 | bait | n04182152 | shadow_box |
| | | | *Continued on next page* |

| Synset ID | Class Name | Synset ID | Class Name |
|-----------|------------|-----------|------------|
| n02783994 | baluster | n04184435 | shaper |
| n02786331 | bandbox | n04186051 | shaving_cream |
| n02799323 | baseball_cap | n04190376 | shelf_bracket |
| n02806379 | bat | n04198722 | shiv |
| n02807523 | bath_oil | n04200258 | shoebox |
| n02807616 | bathrobe | n04200537 | shoehorn |
| n02808185 | bath_salts | n04206356 | shotgun |
| n02811618 | battle_cruiser | n04210120 | shredder |
| n02812949 | bayonet | n04211219 | shunter |
| n02816656 | beanbag | n04218564 | silencer |
| n02817031 | bearing | n04219424 | silk |
| n02821202 | bedpan | n04221823 | simulator |
| n02823586 | beer_garden | n04224842 | sitar |
| n02826068 | bell_jar | n04228215 | ski_binding |
| n02831237 | beret | n04228693 | ski_cap |
| n02841187 | binnacle | n04233124 | skyscraper |
| n02843553 | bird_feeder | n04237423 | slicer |
| n02851939 | blindfold | n04238321 | slide_fastener |
| n02855089 | blower | n04248851 | snare |
| n02868975 | bone-ash_cup | n04252331 | snowshoe |
| n02869249 | bones | n04252653 | snow_thrower |
| n02874442 | bootjack | n04253057 | snuffbox |
| n02877266 | bottle | n04258333 | solar_heater |
| n02879087 | bouquet | n04258859 | soldering_iron |
| n02880842 | Bowie_knife | n04260364 | sonogram |
| n02881757 | bowler_hat | n04269270 | spark_plug |
| n02882190 | bowling_alley | n04270891 | spear |
| n02882301 | bowling_ball | n04272389 | spectator_pump |
| n02882647 | bowling_pin | n04273285 | speculum |
| n02887489 | brace | n04282494 | splint |
| n02890940 | brake_shoe | n04284869 | sport_kite |
| n02892948 | brass_knucks | n04287747 | spray_gun |
| n02893608 | breadbasket | n04289027 | sprinkler |
| n02893941 | bread_knife | n04290259 | spur |
| n02895438 | breathalyzer | n04292921 | squeegee |
| n02903204 | broadcaster | n04303357 | staple_gun |
| n02904803 | brocade | n04303497 | stapler |
| n02905036 | broiler | n04306592 | stator |
| n02910145 | bucket_seat | n04314914 | step |
| n02911332 | buffer | n04315342 | step-down_transformer |
| n02925009 | bushel_basket | n04320973 | stirrup |
| n02927764 | butter_dish | n04326896 | stool |
| n02928608 | button | n04331639 | straightener |
| n02939866 | caliper | n04335886 | streetlight |
| n02940385 | call_center | n04344003 | stud_finder |
| n02944579 | camouflage | n04346157 | stun_gun |
| n02947660 | canal_boat | n04358117 | supercomputer |
| n02951703 | canopic_jar | n04364160 | surge_suppressor |
| n02952237 | canopy | n04364545 | surgical_instrument |
| n02955247 | capacitor | n04373894 | sword |
| n02956699 | capitol | n04386051 | tailstock |
| n02957008 | capote | n04387095 | tam |
| n02960690 | carabiner | n04389854 | tank_engine |
| n02960903 | carafe | n04392113 | tape |
| n02962061 | carboy | n04409128 | tender |
| n02962200 | carburetor | n04414675 | Tesla_coil |
| n02967782 | carpet_sweeper | n04417180 | textile_machine |
| | | | *Continued on next page* |

| Synset ID | Class Name | Synset ID | Class Name |
|-----------|------------|-----------|------------|
| n02970685 | car_seat | n04419073 | theodolite |
| n02973017 | cartridge_holder | n04421872 | thermometer |
| n02973904 | carving_knife | n04432662 | ticking |
| n02976249 | case_knife | n04438897 | tin |
| n02977330 | cashmere | n04442441 | toaster_oven |
| n02978055 | casket | n04449966 | tomahawk |
| n02981024 | catacomb | n04450749 | tongs |
| n02982232 | catapult | n04451318 | tongue_depressor |
| n02986160 | cattle_guard | n04452528 | tool_bag |
| n02988066 | C-clamp | n04453156 | toothbrush |
| n02993194 | cenotaph | n04453390 | toothpick |
| n02993368 | censer | n04469514 | trampoline |
| n02998003 | cereal_box | n04474035 | transporter |
| n03005033 | chancellery | n04477387 | treadmill |
| n03005285 | chandelier | n04479939 | trestle_bridge |
| n03011355 | checker | n04482177 | tricorn |
| n03014440 | chessman | n04483073 | trigger |
| n03019685 | chin_rest | n04483925 | trimmer |
| n03027250 | chuck | n04488202 | trophy_case |
| n03029445 | churn | n04489817 | trowel |
| n03030353 | cigar_box | n04495698 | tudung |
| n03033362 | circuit | n04495843 | tugboat |
| n03034405 | circuitry | n04496872 | tumbler |
| n03041114 | cleat | n04497801 | tuning_fork |
| n03049924 | cloth_cap | n04502502 | tweed |
| n03050655 | clothes_dryer | n04502851 | twenty-two |
| n03051249 | clothespin | n04520784 | vane |
| n03061674 | cockpit | n04525821 | Venn_diagram |
| n03063199 | coffee_filter | n04526964 | ventilator |
| n03064758 | coffin | n04532831 | vibraphone |
| n03066359 | coil_spring | n04533199 | vibrator |
| n03067093 | cold_cathode | n04538552 | vise |
| n03075097 | comb | n04540255 | volleyball_net |
| n03080633 | compass | n04554406 | washboard |
| n03082807 | compressor | n04556408 | watch_cap |
| n03087069 | concrete_mixer | n04559166 | water_cooler |
| n03089753 | conference_center | n04568069 | weathervane |
| n03097362 | control_center | n04568841 | webbing |
| n03099147 | convector | n04579056 | whiskey_bottle |
| n03101664 | cookie_jar | n04582869 | wicket |
| n03103563 | coonskin_cap | n04586581 | winder |
| n03105088 | copyholder | n04589325 | window_box |
| n03105467 | corbel | n04590746 | windshield_wiper |
| n03115400 | cotton_flannel | n04594489 | wire |
| n03119510 | coupling | n04606574 | wrench |
| n03122073 | covered_bridge | n04613939 | Zamboni |
| n03133415 | crock | n04615226 | zither |
| n03140431 | cruet | n06275095 | cable |
| n03141702 | crusher | n07579688 | piece_de_resistance |
| n03150232 | curler | n07579917 | adobo |
| n03156767 | cylinder_lock | n07580359 | casserole |
| n03158885 | dagger | n07591961 | paella |
| n03161450 | damper | n07593471 | viand |
| n03175457 | densitometer | n07594066 | cake_mix |
| n03176386 | denture | n07607138 | chocolate_kiss |
| n03176594 | deodorant | n07611991 | mousse |
| n03180504 | destroyer | n07613815 | jello |

*Continued on next page*

| Synset ID | Class Name | Synset ID | Class Name |
|---|---|---|---|
| n03219135 | doll | n07617708 | plum_pudding |
| n03229244 | dowel | n07617932 | corn_pudding |
| n03232543 | drain_basket | n07618119 | duff |
| n03233744 | drawbridge | n07618432 | chocolate_pudding |
| n03235327 | drawknife | n07621618 | garnish |
| n03235796 | drawstring_bag | n07624466 | turnover |
| n03240892 | drill_press | n07641928 | fish_cake |
| n03247083 | dropper | n07642361 | fish_stick |
| n03249342 | drugstore | n07648913 | buffalo_wing |
| n03250279 | drumhead | n07648997 | barbecued_wing |
| n03253796 | duffel | n07654148 | barbecue |
| n03254046 | duffel_coat | n07654298 | biryani |
| n03254862 | dulcimer | n07655263 | saute |
| n03255899 | dumpcart | n07665438 | veal_parmesan |
| n03256166 | dump_truck | n07666176 | veal_cordon_bleu |
| n03261776 | earphone | n07680313 | bap |
| n03266371 | eggbeater | n07680517 | breadstick |
| n03267468 | ejection_seat | n07680761 | brown_bread |
| n03272125 | electric_hammer | n07681450 | challah |
| n03282295 | embassy | n07681691 | cinnamon_bread |
| n03287351 | energizer | n07682197 | crouton |
| n03293741 | equalizer | n07682316 | dark_bread |
| n03296081 | escapement | n07682477 | English_muffin |
| n03309356 | eyepatch | n07682624 | flatbread |
| n03326795 | felt | n07682808 | garlic_bread |
| n03329663 | ferry | n07684164 | matzo |
| n03331077 | fez | n07684517 | raisin_bread |
| n03342127 | finger-painting | n07685730 | rye_bread |
| n03345837 | fire_extinguisher | n07686720 | sour_bread |
| n03350204 | fishbowl | n07686873 | toast |
| n03351434 | fishing_gear | n07687053 | wafer |
| n03356982 | flannel | n07696527 | butty |
| n03359566 | flask | n07696625 | ham_sandwich |
| n03363749 | flintlock | n07696728 | chicken_sandwich |
| n03364599 | float | n07696839 | club_sandwich |
| n03367410 | florist | n07696977 | open-face_sandwich |
| n03367545 | floss | n07697699 | Sloppy_Joe |
| n03378174 | food_processor | n07697825 | bomber |
| n03379828 | footbridge | n07698250 | gyro |
| n03392741 | franking_machine | n07698401 | bacon-lettuce-tomato_sandwich |
| n03397266 | frigate | n07698543 | Reuben |
| n03397947 | Frisbee | n07698672 | western |
| n03398228 | frock_coat | n07698782 | wrap |
| n03407369 | fuse | n07708124 | julienne |
| n03420801 | garrison_cap | n07709172 | potherb |
| n03423479 | gas_heater | n07713267 | pieplant |
| n03423719 | gasket | n07713763 | mustard |
| n03429288 | gauge | n07715221 | brussels_sprouts |
| n03430418 | gazebo | n07723039 | leek |
| n03431745 | gearing | n07730406 | celery |
| n03432129 | gearshift | n07733394 | gumbo |
| n03440682 | glockenspiel | n07750736 | Jordan_almond |
| n03442756 | goal | n07750872 | apricot |
| n03448590 | gorget | n07753743 | passion_fruit |
| n03451711 | graduated_cylinder | n07755411 | melon |
| n03456024 | gravy_boat | n07758680 | grape |
| n03456665 | greatcoat | n07762114 | papaw |
| | | | *Continued on next page* |

| Synset ID | Class Name | Synset ID | Class Name |
|-----------|-----------|-----------|-----------|
| n03460040 | grinder | n07762740 | ackee |
| n03466493 | guided_missile_cruiser | n07765073 | date |
| n03469903 | gunnysack | n07765999 | jujube |
| n03475823 | hairdressing | n07770763 | pumpkin_seed |
| n03490119 | hand_truck | n07775197 | sunflower_seed |
| n03490884 | hanger | n07806221 | salad |
| n03494537 | harmonium | n07817871 | fennel |
| n03495039 | harness | n07823951 | curry |
| n03497352 | hasp | n07830593 | hot_sauce |
| n03505133 | headrest | n07832416 | pesto |
| n03505504 | headscarf | n07835457 | hollandaise |
| n03506184 | headstock | n07835921 | bourguignon |
| n03506727 | hearing_aid | n07837362 | white_sauce |
| n03507241 | hearth | n07838073 | gravy |
| n03542333 | hotel | n07840027 | veloute |
| n03542605 | hotel-casino | n07841495 | boiled_egg |
| n03548402 | hula-hoop | n07842202 | poached_egg |
| n03549473 | hunting_knife | n07842308 | scrambled_eggs |
| n03553019 | hydrofoil | n07842433 | deviled_egg |
| n03565288 | imprint | n07842605 | shirred_egg |
| n03571625 | ink_bottle | n07842753 | omelet |
| n03572107 | inkle | n07843464 | souffle |
| n03572321 | inkwell | n07843636 | fried_egg |
| n03589513 | jack | n07849336 | yogurt |
| n03609397 | kazoo | n07861557 | coq_au_vin |
| n03610682 | kepi | n07861813 | chicken_and_rice |
| n03612965 | kettle | n07862244 | bacon_and_eggs |
| n03614782 | keyhole | n07862348 | barbecued_spareribs |
| n03615790 | khukuri | n07862461 | beef_Bourguignonne |
| n03625355 | knit | n07862611 | beef_Wellington |
| n03626760 | knocker | n07864756 | chicken_Kiev |
| n03628215 | koto | n07864934 | chili |
| n03631177 | lace | n07865196 | chop_suey |
| n03641569 | lanyard | n07865484 | chow_mein |
| n03645011 | latch | n07866015 | croquette |
| n03659809 | lever | n07866151 | cottage_pie |
| n03659950 | lever_lock | n07866277 | rissole |
| n03668067 | lightning_rod | n07866409 | dolmas |
| n03680858 | lobster_pot | n07866723 | egg_roll |
| n03699591 | machete | n07866868 | eggs_Benedict |
| n03718212 | man-of-war | n07867021 | enchilada |
| n03719743 | mantilla | n07867164 | falafel |
| n03720163 | map | n07867324 | fish_and_chips |
| n03721047 | marble | n07867421 | fondue |
| n03725600 | Mason_jar | n07868200 | French_toast |
| n03742019 | medicine_ball | n07868340 | fried_rice |
| n03743279 | megaphone | n07868508 | frittata |
| n03760944 | microtome | n07868830 | galantine |
| n03765561 | mill | n07868955 | gefilte_fish |
| n03767112 | millstone | n07869522 | corned_beef_hash |
| n03774327 | miter_box | n07869611 | jambalaya |
| n03775388 | mixer | n07869775 | kabob |
| n03784270 | monstrance | n07870313 | seafood_Newburg |
| n03789946 | motor | n07871436 | meatball |
| n03795269 | mouthpiece | n07872593 | moussaka |
| n03797182 | muffler | n07873464 | pilaf |
| n03802007 | musket | n07874780 | porridge |
| | | | *Continued on next page* |

| Synset ID | Class Name | Synset ID | Class Name |
|---|---|---|---|
| n03805280 | nailfile | n07875436 | risotto |
| n03814817 | neckerchief | n07876651 | Scotch_egg |
| n03819448 | nest_egg | n07877299 | Spanish_rice |
| n03820318 | net | n07877675 | steak_tartare |
| n03836451 | nut_and_bolt | n07877849 | pepper_steak |
| n03839671 | observatory | n07878647 | stuffed_peppers |
| n03840823 | octant | n07878926 | stuffed_tomato |
| n03844045 | oil_lamp | n07879072 | succotash |
| n03858418 | ottoman | n07879174 | sukiyaki |
| n03865557 | overpass | n07879350 | sashimi |
| n03875955 | paintball_gun | n07879450 | sushi |
| n03883524 | pannier | n07879659 | tamale |
| n03884926 | pantheon | n07879953 | tempura |
| n03887185 | paper_fastener | n07880213 | terrine |
| n03890093 | parer | n07880325 | Welsh_rarebit |
| n03890514 | pari-mutuel_machine | n07880458 | schnitzel |
| n03897943 | patch | n07880751 | taco |
| n03901229 | pavior | n07881404 | tostada |
| n03904909 | peeler | n07929351 | coffee_bean |
| n03919430 | pestle | n07933154 | tea_bag |
| n03920641 | pet_shop | n07937461 | couscous |
| n03923379 | phial | n07938149 | vitamin_pill |
| n03923918 | phonograph_needle | n09451237 | supernova |
| n03936466 | pile_driver | n09818022 | astronaut |
| n03937931 | pillion | n09834699 | ballet_dancer |
| n03938037 | pillory | n09846755 | beekeeper |
| n03938401 | pillow_block | n09913593 | cheerleader |
| n03939178 | pilot_boat | n10091651 | fireman |
| n03941231 | pinata | n10366966 | nurse |
| n03941417 | pinball_machine | n10514429 | referee |
| n03941684 | pincer | n10521662 | reporter |
| n03946076 | pipe_cutter | n10772092 | weatherman |
| n03948950 | piston | n11706761 | avocado |
| n03952576 | pizzeria | n11851578 | prickly_pear |
| n03955489 | plane_seat | n11877283 | kohlrabi |
| n03967396 | plotter | n11879054 | bok_choy |
| n03968293 | plug | n12088223 | yam |
| n03968581 | plughole | n12136392 | rattan |
| n03973628 | pocketknife | n12158031 | gourd |
| n03977592 | police_boat | n12158443 | pumpkin |
| n03983612 | poplin | n12172364 | okra |
| n03993180 | pouch | n12246232 | blueberry |
| n03996416 | power_shovel | n12301445 | olive |
| n04000311 | press | n12333771 | guava |
| n04011827 | propeller | n12352990 | plantain |
| n04013729 | prosthesis | n12373100 | papaya |
| n04015908 | protractor | n12399132 | mulberry |
| n04016846 | psaltery | n12400489 | breadfruit |
| n04020298 | pulley | n12433081 | onion |
| n04022332 | pump | n12441183 | asparagus |
| n04024862 | punnet | n12501202 | tamarind |
| n04039848 | radar | n12515925 | chickpea |
| n04041243 | radiator_cap | n12544539 | lentil |
| n04043411 | radio-phonograph | n12560282 | pea |
| n04049753 | rain_stick | n12578916 | cowpea |
| n04050933 | ramekin | n12636224 | medlar |
| n04051549 | ramp | n12638218 | plum |

*Continued on next page*

| Synset ID | Class Name | Synset ID | Class Name |
|-----------|------------|-----------|------------|
| n04054670 | rasp | n12642090 | wild_cherry |
| n04063154 | record_changer | n12648045 | peach |
| n04064401 | record_player | n12709103 | pomelo |
| n04071263 | regalia | n12709688 | grapefruit |
| n04072960 | relay | n12711984 | lime |
| n04075291 | remote_terminal | n12713063 | kumquat |
| n04075916 | repair_shop | n12744387 | litchi |
| n04079933 | resistor | n12761284 | mango |
| n04082562 | retainer | n12771192 | persimmon |
| n04093625 | rink | n12805146 | currant |
| n04093775 | riot_gun | n12911673 | tomatillo |
| n04095577 | riveting_machine | n13136316 | bean |
| n04097760 | roaster | n13136556 | nut |
| n04098513 | rocker | | |

