# OpenReview forum: "ImageNet-OOD: Deciphering Modern Out-of-Distribution Detection Algorithms"
_ICLR.cc/2024/Conference — ICLR 2024 poster_

### Official Review · Reviewer_pNua · 2023-10-30

**Soundness:** 3 good
**Presentation:** 3 good
**Contribution:** 3 good
**Rating:** 8
**Confidence:** 5

**Summary:**

This work introduces a new OOD detection benchmark for ImageNet-1K, namely ImageNet-OOD (IN-OOD for short). Compared to existing benchmarks (e.g., Species, OpenImage-O, SSB, NINCO), IN-OOD minimizes covariate shifts and operates at a large scale (in terms of # OOD categories and # images). With this new benchmark, the authors evaluate several recent post-hoc detectors under both 1) novel class detection and 2) failure detection schemes. The major finding is that existing detectors are (undesirably) much more sensitive to covariate shifts than semantic shifts, and most detectors do not provide practical benefits over the simplest baseline MSP.

**Strengths:**

1. Although not for the first time, this work does focus on and point out several crucial issues for OOD detection research (which unfortunately still haven't been paid enough attention by the researchers in this field). These issues include: 1) the lack of a semantic shift-only, clean, and large-scale OOD dataset for ImageNet-1K, 2) the sensitivity of existing methods to covariate shift, and 3) the mismatch between a) the ultimate goal of OOD detection and b) the "wrong" goal reflected by the current evaluation which is the result of issue 1) and 2).
2. The constructed IN-OOD would indeed be valuable, especially considering the rigorousness introduced by the several filtering processes and the final human inspection. I could imagine how big the human efforts involved in this process are, and I personally appreciate it.
3. Performing evaluations under both new-class detection and failure detection setting is good, which can provide a unified assessment.

**Weaknesses:**

1. An important reference, OpenOOD v1.5 [1] is missing (released on arXiv in June 2023). Their evaluation results in the full-spectrum OOD detection setting (considering semantic-shifted and covariate-shifted samples together [2]) are also concrete evidences that current OOD detection methods, not restricting to post-hoc methods, are very sensitive to covariate shifts. This is actually presented as one of their major observations, and thus I believe it is necessary to discuss this work at least in the Related Work section.

2. Like I said, I appreciate the efforts in constructing IN-OOD and I recognize the value in this new dataset. However, I wouldn't say that the observation of "OOD detectors are more sensitive to covariate shifts than semantic shifts", which is one of the claimed contributions, is new. Evidences include both [1, 2]. Another major observation of this work, "the practical benefits of these algorithms disappear under new-class detection", is also similar to one presented in [1], where they find the improvements in "near-OOD" (which essentially has less covariate shifts than "far-OOD") detection is limited.

3. This is not really a weakness. I like the example in Figure 1 where "Animal, Vehicle" are ID, and training images of "Animal" is dog while test images could be cat. This actually points to a type of covariate shift called Subpopulation Shift [3]. I think the discussion on this example could be made more clear by explicitly discussing the relationship between OOD detection and Subpopulation Shift (although I agree that this could often times be application-dependent).

4. Lastly, again this is not technically a weakness, but part of me feels that this work might suit dedicated dataset & benchmark track better.

[1] OpenOOD v1.5: Enhanced Benchmark for Out-of-Distribution Detection

[2] Full-Spectrum Out-of-Distribution Detection

[3] BREEDS: Benchmarks for Subpopulation Shift

**Questions:**

As demonstrated by NINCO work, human inspection is necessary for constructing clean OOD datasets (which is also recognized in this work). However, human inspection could be extremely costly, which I believe is the reason why NINCO itself is limited in size (a few hundreds or at most thousands of images). IN-OOD in comparison has a total of 31,807 images. I was wondering how thorough and rigorous the human inspection was for the "final review" of IN-OOD at this size. How many human inspectors were involved and how long did it take for the final review?

---

> ### Author Response · Authors · 2023-11-20
>
> Dear Reviewer,
>    Thank you for giving us your time and energy for the thorough and insightful review of our paper. We are encouraged that you appreciate the identification of crucial issues in OOD detection research, their mitigation in our novel dataset, and our unified assessment of OOD detection algorithms. We greatly appreciate the references and concerns you pointed out and updated parts of the paper to improve it. Please see our response to the concerns below:
>
>
> **W1-2: OpenOOD v1.5 shown that OOD detectors are susceptible towards covariate shifts and practical benefits disappear in near-OOD datasets**
>
> Thank you for pointing this out. We updated the related works section (section A of appendix) to include a pointer to OpenOOD v1.5 (blue text on pg. 14). The key distinction with our analysis is that we studied the effect of covariate shift independently and found that the baseline MSP is more robust towards covariate shifts than modern OOD detection algorithms. OpenOOD v1.5 performed the analysis in the full spectrum (ID combined with covariate shift) setting, which hid this fact, and only concluded that such setting poses a challenge for OOD detection. We updated section 4.3 and introduction to make this point more clear. Additionally, we would like to note that as you point out, this work has only appeared on arxiv and has not yet been published in peer-reviewed venues, thus we hope it does not affect assessment of novelty of our work as per ICLR guidelines.
>
>
> **W3: Animal and Vehicle is an example of subpopulation shift**
>
> This is a great suggestion and we updated the paper to reflect this in the “semantically-grounded covariate shifts” paragraph of Section 3.2 (bottom of pg. 4, highlighted in blue).
>
>
> **W4: paper is better suited for Dataset and Benchmark**
>
> We understand where you’re coming from but respectfully disagree. The Datasets and Benchmarks track provides a much-needed venue to share generalizable insights into dataset construction and present innovative approaches to benchmarking. In contrast, in our work, while the ImageNet-OOD dataset is very important to our analysis and is carefully constructed to suit this need, it does not necessarily reveal meaningful insights into dataset construction in general. Further, we believe our analyses and takeaways about OOD detection algorithms are critical towards guiding better methodological design; the construction of the dataset itself is a secondary contribution that merely enables these findings. This is why we maintain that this paper is better suited  for a broad audience conference instead of a dedicated track.
>
>
> **Q1: how thorough and rigorous is the human inspection and how does it compare to NINCO?**
>
> Two rounds of manual human inspection are performed by the authors of the paper, first at the class level (20 hours), then at the image level (6 hours). We first manually inspected sample images from direct neighbors of ImageNet-1K classes and selected 1000 classes that are visually unambiguous to their ImageNet-1K neighbors. Then, we inspect the top 1000 images that are closest to ImageNet-1K in terms of ResNet-50 feature distance to filter out classes that are visually ambiguous with non-neighboring ImageNet-1K classes to arrive at the final 637 classes. Since the ImageNet-21K images are all validated by crowd workers from Amazon Mechanical Turk, we dedicated most of our time to class-level inspection, while still performing image-level analysis as an additional layer of reassurance. Please refer to Appendix Section H for more details on the construction of ImageNet-OOD.
>
>
> NINCO was a smaller dataset designed to avoid data contamination of in-distribution classes. In comparison, in addition to avoiding data contamination, our dataset is designed to mitigate unwanted covariate shifts, which is crucial for our analysis.

---

> > ### Comment · Reviewer_pNua · 2023-11-20
> > **Thanks for the rebuttal**
> >
> > I have read the responses from the authors, including those to other reviewers (especially JbD1), and my questions / concerns have all been addressed. One further comment. I suggest to include some descriptions in the answer to Q1 into "Final Class Selection" in Sec. 3.2 (e.g., the time cost of human inspection). Also, I don't really find where in the main text it points to Appendix H, which definitely should be added.
> >
> > Overall, like I said before, the constructed dataset IN-OOD (which is of large size, minimizes covariate shift, and is "clean" / unambiguous) is valuable for the field (the closest alternative, NINCO, is limited in size). I'm thus raising my score to 8.

---

> > > ### Author Response · Authors · 2023-11-21
> > >
> > > Dear Reviewer,
> > >
> > > We have updated the paper to address the two comments. We would like to thank you again for the valuable feedback.

---

### Official Review · Reviewer_YNoR · 2023-10-30

**Soundness:** 2 fair
**Presentation:** 2 fair
**Contribution:** 3 good
**Rating:** 6
**Confidence:** 3

**Summary:**

This paper introduces ImageNet-OOD, a new out-of-distribution (OOD) detection dataset that minimizes covariate shift compared to ImageNet-1K by manually selecting semantically different classes from ImageNet-21K. Using this dataset, the authors evaluate several recent OOD detection algorithms and find they offer little benefit over the maximum softmax probability (MSP) baseline for detecting semantic shifts. The key conclusions are:
1. Modern OOD detectors are much more sensitive to covariate shifts than semantic shifts.
2. On ImageNet-OOD, modern OOD detectors offer <1% AUROC improvement over MSP for new-class detection.
3. For failure detection, MSP still outperforms modern OOD detectors on ImageNet-OOD.
4. The benefits of modern OOD detectors come more from better separating incorrect in-distribution examples rather than improving on semantic shifts.

**Strengths:**

1. ImageNet-OOD appears to be a useful benchmark for evaluating OOD detection methods on semantic shifts. The careful data curation process is commendable.
2. The analyses on the susceptibility of modern methods to covariate shifts are insightful. The experiments are comprehensive across different datasets, algorithms, and metrics.
3. The finding that MSP remains a strong baseline is an important result for the OOD detection community. It helps calibrate expectations on recent progress.

**Weaknesses:**

1. While covariate shift robustness is desirable, the goal of semantic shift detection is also useful in many applications like open set recognition. The heavy focus on covariate shifts undervalues semantic shift detection.
2. More analysis could be provided on the characteristics of examples that lead methods to confuse covariate and semantic shifts.
The writing and organization needs polish in some areas. The high-level conclusions could be stated more clearly in the intro and abstract.
3. In summary, this is an reasonable contribution introducing a new dataset and providing useful experiments analyzing modern OOD detection methods. I suggest acceptance after revisions to clarify the presentation and provide additional analysis/discussion.

**Questions:**

No more questions.

---

> ### Author Response · Authors · 2023-11-20
>
> Dear Reviewer,
>    Thank you for giving us your time and energy for the thorough and insightful review of our paper. We are encouraged that you appreciate the usefulness of our novel dataset, insightful analysis towards susceptibility towards covariate shift, and calibration of expectations in the OOD community. We greatly appreciate the confusions and weaknesses you pointed out and updated parts of the paper to improve it. Please see below:
>
>
> **W1: focus on covariate shift undervalues semantic shift**
>
> One of the primary goals of our work is to actually curate a clean dataset to better evaluate new-class detection.  We directly emphasize the motivation of constructing ImageNet-OOD at the start of Section 3: “...new-class detection is still relevant in practice with the growing interest in adaptive learning systems.” Additionally, we delineate the difference between new-class detection and failure detection at the end of section 2 to emphasize the importance of semantic shift. We made some minor changes at the beginning of section 3 (highlighted in blue) to make this point clearer.
>
>
> **W2: analysis on characteristics of examples that lead methods to confuse covariate and semantic shift**
>
> This is a great suggestion. We conducted additional qualitative analysis (included in Section G of the appendix) and found that the OOD detection algorithm ViM tends to score texture-like images as more OOD when compared to MSP and the OOD detection algorithm ASH-b tends to score text-like images more OOD when compared to MSP. Additionally, we observe that text-like/texture-like images from ImageNet-OOD are more closely aligned with the domain of ImageNet-1K. Since ASH-b and ViM are the two best performing algorithms on OpenImage-O, these visual examples explain the disappearing performance gains when evaluating on ImageNet-OOD.
>
>
> **W3: presentation and writing issues**
>
> Thank you for pointing this out. We clarified our high level contribution at the end of our introduction (highlighted in blue). Please note that ICLR policy this year prevents us from making changes to the abstract. We also did another pass to fix some minor grammar and presentation issues (highlighted in gray).

---

### Official Review · Reviewer_PDzS · 2023-10-30

**Soundness:** 3 good
**Presentation:** 3 good
**Contribution:** 2 fair
**Rating:** 6
**Confidence:** 3

**Summary:**

This paper introduces a new out-of-distribution (OOD) task dataset, primarily composed of data from ImageNet-1K and ImageNet-21K. The authors selected a portion of the data using specific rules and meticulous manual annotation for OOD tasks. The authors discovered that many state-of-the-art (SOTA) methods performed worse than certain baseline methods on this dataset, prompting further reflection on OOD tasks.

**Strengths:**

1. The motivation of this article is meaningful. Currently, available OOD detection datasets do have certain issues, and by using a more carefully selected dataset, it is possible to define the problem of OOD detection better.

2. The dataset proposed in this article has inspired the development of OOD detection tasks. The authors discovered that many state-of-the-art methods did not perform well on their dataset, and based on this, they made their findings, which are beneficial for further research.

3. The description in this article is clear, making it easy for readers to understand the characteristics of the dataset, its construction method, experimental results, and conclusions.

**Weaknesses:**

1. The focus of this paper is primarily on the dataset and analysis, which are undoubtedly meaningful aspects. However, the author fails to provide their own methods to improve the effectiveness of OOD detection tasks, which results in a lack of depth and contribution in this paper.

2. The author's comparison methods lack some of the latest approaches. In recent conferences such as CVPR 2023, new methods have been proposed. Including these methods in the comparison would make the article more comprehensive. Additionally, many methods may be sensitive to hyperparameters, so it would be beneficial to discuss the adjustment of hyperparameters when changing to new datasets.

**Questions:**

As shown in the weakness.

---

> ### Author Response · Authors · 2023-11-20
>
> Dear Reviewer,
>    Thank you for giving us your time and energy for the thorough and insightful review of our paper. We are encouraged that you appreciate the importance of our motivation, the benefits of our novel dataset, and the clear presentation of method and results. We greatly appreciate the suggestion provided. Please see our responses to the weaknesses you pointed out below:
>
>
> **W1: paper focuses on dataset and analysis but does not provide its own OOD detection method**
>
> Absolutely, we do not provide a new method for OOD detection. As we note in the abstract, our contributions are as follows: “we present ImageNet-OOD, a clean semantic shift dataset that minimizes the interference of covariate shift. Through comprehensive experiments, we show that OOD detectors are more sensitive to covariate shift than to semantic shift, and the benefits of recent OOD detection algorithms on semantic shift detection are minimal. Our dataset and analyses provide important insights for guiding the design of future OOD detectors.”
> Our work follows a long line of works (please see section A of appendix) which provide detailed analyses and insights into existing paradigms and approaches, rather than developing new models.
>
>
> **W2: lack of new methods from CVPR and lack of hyperparameter tuning**
>
> Integrating recent CVPR methods is an excellent suggestion. We added a new method Max-Cosine (Zhang and Xiang) from CVPR 2023 and also another method ASH-B (Djurisic et al.) from ICLR 2023. We kindly ask that you please note that works published in CVPR 2023 are within the four months prior to ICLR submission and therefore, are considered contemporaneous by the ICLR policy.
>
>
> We included Max-Cosine and ASH-B in Table 1 (pg 8), integrated them into the summary of the results in Section 4.3, and added additional qualitative analysis in Section G of the appendix (all highlighted in blue).  Briefly, we find the same conclusion: the best performing detector is now ASH-B on OpenImage-O with improvement of 5.5 points on AUROC while the detector perform worse than the MSP baseline on ImageNet-OOD; Max-Cosine is the best detector on ImageNet-OOD but the improvement from the MSP baseline is still less than 1 point on AUROC.
>
>
> Regarding the hyperparameters, the methods we used are either hyperparameter free (MSP, Energy, Max-Logit, Mahalanobis, and the newly added Max-Cosine) or the methods are tuned towards ImageNet, our particular setup,  in the original paper of the method (ViM, KNN, ReAct, and newly added ASH-B). As a result, for fair evaluation, we used the best suggested hyperparameter based on the ablation studies performed in the original paper. We added this clarification at the end of the second paragraph of section 4.

---

### Official Review · Reviewer_JbD1 · 2023-10-31

**Soundness:** 2 fair
**Presentation:** 3 good
**Contribution:** 2 fair
**Rating:** 6
**Confidence:** 4

**Summary:**

This paper analyzes the capabilities of current OOD detection algorithms highlighting a bias towards covariate shifts. In consequence, the authors introduce ImageNet-OOD, a dataset that can assess the impact of semantic shifts without the influence of covariate shifts.

The authors address the limitation of current benchmarks to correctly evaluate out-of-distribution (OOD) detection algorithms due to a missing clear separation between semantic and covariate shifts.

For the proposed set, the corresponding in-distribution (ID) dataset is considered to be ImageNet-1K.
ImageNet-OOD is a manually curated dataset, with 31807 images from 637 classes.
When building ImageNet-OOD, the authors started from ImageNet-21K and curated it in order to address the following issues:
   - semantic ambiguity
   - visual ambiguity
   - unnecessary covariate shifts

Curation steps:
   - **1 All ImageNet-1K classes, their hypernyms, and hyponyms** - remove classes corresponding to ImageNet-1k classes, their hypernyms and hyponyms
   - **2 Hyponyms of "Organism"** - as there is an issue with the classification of natural beings in WordNet (classified by both technical biological levels and non-technical categories), all classes having 'organism' as a hyponym are removed
   - **3 Semantically-grounded covariate shifts** - remove potential 'semantically-grounded covariate shifts'. If we train a binary classifier to differentiate between 'dog' and 'vehicle', it can also be understood as an 'animal' vs. 'vehicle' classifier => class 'cat' can be considered as a semantically-grounded covariate shift  and should be removed to avoid confusions. Considering each pair of classes from ImageNet-1K, the common ancestor is identified, and the classes are associated with the immediate descendants of this common ancestor (most general description). Further, exclude all classes from ImageNet-21K that are hyponyms of the general descriptions identified above
   - **4 Final Class Selection** - remove samples generating semantic ambiguity due to inaccurate hierarchical relations in ImageNet labels (e.g. violin and viola)

The authors perform extensive experiments on seven OOD detection algorithms across 13 network architectures highlighting that:
 - OOD detection algorithms are more sensitive to covariate shifts than semantic shifts
- the practical benefits of new algorithms vs. MSP (maximum softmax probability) disappear under both new-class detection and failure detection scenarios

**Strengths:**

**S1** Introduce ImageNet-OOD, a curated OOD dataset for ImageNet-1K.

**S2** Highlighting the ambiguity between semantic and covariate shifts for the OOD detection problem.

**S3** Extensive experimental analysis, considering 7 OOD detection algorithms and 13 model architectures.

**Weaknesses:**

**W1** Introducing separate datasets for assessing semantic and covariate shifts is relevant for roughly understanding OOD detection capabilities.
Yet, in a real-case scenario, both semantic and covariate shifts will be present, and whether we wish to ignore one of the two should be specified through the training set (e.g. using a multi-environment setup - see [1])
If both semantic and covariate shifts are to be considered, then it is expected that the OOD detection algorithm will first identify the one generating the highest shift.
(see question **Q1**)

**W2** The curation steps can benefit from an in-depth analysis
See questions **Q2**, **Q3**, **Q4** and **Q5**

**W3** Conclusion of Sec.4.2, where randomly initialized models are considered for testing OOD detectors. Here, the authors conclude that there is a bias towards detecting covariate shifts even for untrained models.
This observed bias is most probably justified by the inductive bias of the considered ResNet-50 model.
Randomly initialized models are more sensitive to specific covariate shifts, but this observation cannot be generalized to any covariate shift. (see question **Q6**)


[1] Smeu et al. Env-aware anomaly detection: Ignore style changes, stay true to content! - NeurIPSW 2022

**Questions:**

**Q1** When evaluating the sensitivity to covariate or semantic shift, apart from the distances towards the closest ImageNet-1K sample (Fig. 3 left), you can also consider distances between the considered datasets (e.g. OTDD [2])
This can help you understand if the sensitivity can be explained by dataset statistics or is simply a model / method bias.
Have you considered such an analysis?
For example, in Table 1, the model trained on ImageNet-1K sees samples from ImageNet-C and ImageNet-OOD. Depending on the distances between those sets and ImageNet-1K, we can understand why covariate or semantic shifts are captured by the OOD detection algorithms.

**Q2**  If we first perform the third curation step "Semantically-grounded covariate shifts", is there any reason to employ step 1 "All ImageNet-1K classes, their hypernyms, and hyponyms"?
By reaching the most general decision boundary for each pair of ImageNet-1K classes and further removing all the classes that fall under those broader decision boundaries from ImageNet-21K you remove both hypernyms and hyponyms.

**Q3** Regarding the example with 'viola' and 'violin' from Figure 1. Is there any reason for this ambiguity to persist after the elimination of 'semantic-grounded covariate shifts'?

**Q4** For the 'Final Class Selection' step, have you considered the implementation of an automated process that exploits, for example, the CLIP embeddings?
Instead of manually searching for those ambiguities, you can use CLIP zero-shot to classify images in both ID and OOD classes and understand potential similarities / confusions.
This would be useful for having a receipt for curating OOD sets based on a considered ID set, without requiring human intervention.

**Q5** Regarding examples from Figure 2: Images with similar visual contents are presented - this means that a certain threshold for visual similarity is considered when removing visually similar classes. How do you choose this threshold, considering that the selection is manually performed. Regarding **Q4**, it would be useful to use such an automatic approach in order to establish a relevant threshold.

**Q6** Regarding Sec. 4.2 - The considered covariate shifts are restricted to image alteration techniques (blur, noise, etc.). But, covariate shifts can also appear when we observe objects in city backgrounds in ID, while in the OOD set, we observe objects on a forest background.

*Q6.1* Have you considered this type of covariate shifts? I assume that the conclusion of this section will not hold in this scenario. Actually, it may be valid for a restricted set of covariate shifts, strongly related with the inductive bias of the considered model architecture.

*Q6.2* Have you performed the same analysis considering semantic shifts? This would be useful in order to conclude that random models are more sensitive to covariate shifts.

[2] Alvarez-Melis and Fusi "Geometric dataset distances via optimal transport" -NeurIPS 2020
[3] Radford et al. Learning transferable visual models from natural language supervision - ICML 2021

---

> ### Author Response · Authors · 2023-11-20
> **Response (1/2)**
>
> Dear Reviewer,
>    Thank you for giving us your time and energy for the thorough and insightful review of our paper. We are encouraged that you appreciate our novel dataset and the extensiveness of our experiments. We've deeply considered your suggestions and are happy to discuss improvements. Please see below:
>
> **W1, Q1: real-world scenarios have both covariate and semantic shift present. Have we considered distance measures between datasets?**
>
> Incorporating dataset distance measurements is a great suggestion and something that we have previously investigated. We tried using the OTDD work you suggested, but unfortunately it was too memory intensive to fit on ImageNet, ImageNet-OOD, ImageNet-R, and ImageNet-Sketch due to the high resolution images and high number of classes (they defined a dataset as feature-label pairs and they only tested on low resolution datasets such as CIFAR-10/Tiny-ImageNet). Instead, we did try a common distance metric from the field of generative modeling: Frechet Inception Distance (FID), which represents the Wasserstein-2 distance in feature space. We did not find a consistent relationship between dataset distance and OOD detection performance. On CLIP embedding, we receive the following FIDs against ImageNet-1K images
> - 1.36 for ImageNet-Sketch
> - 0.99 for ImageNet-R
> - 1.07 for ImageNet-OOD
>
> We observe a similar pattern on FIDs using embeddings of a ResNet-50 model trained on ImageNet-1K against ImageNet-1K images
> - 21.8 for ImageNet-Sketch
> - 13.4 for ImageNet-R
> - 13.9 for ImageNet-OOD
>
> From FID, we observe that ImageNet-R and ImageNet-OOD have similar distances while ImageNet-Sketch is significantly further away. Despite the fact that ImageNet-Sketch is significantly further than ImageNet-R, our results in Table 1 between ImageNet-OOD vs. ImageNet-R/ImageNet-Sketch have similar performance. Thus, we did not find any consistent relationship between dataset distances and OOD detection performance, and therefore, the susceptibility towards covariate shift cannot be explained by dataset distance.
>
> Regarding real-world scenarios and what to do with covariate shift, we build off of existing literature for the proper procedure for handling covariate shift. As outlined in the introduction and described at the end of section 2 under the “modern approach” and “classic approach” subheadings (references provided in more detail in section A of appendix), OOD detection algorithms have two popular use cases: new-class detection and failure detection. The goal of our paper is to provide a dataset that can better assess an OOD detection algorithm’s performance for the task of new-class detection (i.e., in the context of semantic shift). Under the formulation of the task of new-class detection in previous works, covariate shift is not considered OOD, and therefore, should be not detected by an OOD detection algorithm. We added another sentence at the beginning of Section 3 (highlighted in blue) to further clarify.
>
>
> **W2, Q2-Q3: why repeat the hypernym and hyponym removal process for ImageNet-1K classes in addition to most general semantic classes? Why does the violin and viola example exist after removing semantically-grounded shifts?**
>
> You are correct that this process does not need to be repeated for the original ImageNet-1K classes. We divided them in separate sections in the paper for the sake of clarity but we thank you for this suggestion and further clarified in the end of the paragraph describing “semantically-grounded covariate shifts” (Section 3.2, bottom of page 4, highlighted in blue).
>
> Regarding the violin and viola example:  ImageNet-1K contains the classes violin and cello, which share a common parent “bowed stringed instrument”. Therefore, violin class would still remain in the most general decision boundary. As a result, viola would still create issues with the violin class.

---

> ### Author Response · Authors · 2023-11-20
> **Response (2/2)**
>
> **W2, Q4-5: have we considered an automated process such as CLIP zero-shot embeddings to establish a threshold for removing visual similarities?**
>
> This is a great suggestion but we were solving a different issue with our manual class selection. For our use case, we performed manual inspection of the classes and some of their respective images to eliminate visual ambiguity, not identify classes with high visual similarity. Using CLIP zero-shot classification would not be able to distinguish between the two. We illustrate two examples from experimenting with CLIP zero-shot classification on ImageNet-OOD images with label of the image and ImageNet-1k classes:
> - The CLIP model confuses images of Capote (ImageNet-OOD class), which is a type of coat made of wool/fabric with a hood and typically worn by Native Americans, with images of Trench Coat (ImageNet-1K class), which is a long waterproof coat. This is an example of two objects that are visually similar but not visually ambiguous. If we manually inspect the images of Capote and Trench Coat, we can clearly see the subtle but undeniable difference between the objects.
> - The CLIP model confuses dump cart (ImageNet-OOD class) vs. wheelbarrow (ImageNet-1K class). Both objects are visually similar but there is a clear semantic difference between the two objects: one wheel vs. two wheels.
>
> Visual ambiguity, in contrast to visual similarity, arises due to flaws in the data annotation process of the original ImageNet-21K dataset rather than properties of the data itself. In the ImageNet-21K dataset, data labels are derived from queries to the internet search engine and verified using human annotators from Amazon Mechanical Turk. They ask human annotators the question “is X in the image?” based on the query used. This can create labeling mistakes from visually ambiguous objects such as the violin/viola example shown in Figure 1. Given a picture of viola without any context clues to its size, it is difficult for a human to distinguish the image from violin vs viola. Since the human annotators were asked “is this a violin” or “is this a viola” instead of contrasting between the two, the labeling can go either way depending which question was asked. We circumvent this issue with manual inspection of the classes using our prior knowledge and intuition about object properties to select 637 classes for ImageNet-OOD that would not have such an issue. This way we can be a lot more confident we did not inadvertently include images potentially corresponding to ImageNet-1K classes in the ImageNet-OOD set.
>
> Reference: Section 3 of Deng, Jia, et al. "ImageNet: A large-scale hierarchical image database." CVPR 2009
>
>
>
> **W3, Q6: have we considered other types of covariate shifts on random models? How does semantic shift compare?**
>
> This is a great question. We actually considered such covariate shifts in Figure 4 of the paper with ImageNet-R (https://github.com/hendrycks/imagenet-r). ImageNet-R is a dataset consisting of ImageNet-1K objects in different domains such as video games, paintings, etc.
> We did not include background shifts as such datasets do not currently exist for ImageNet-1K to the best of our knowledge and are difficult to curate. This is because ImageNet-1K, which is the most commonly-used in-distribution dataset in the field, is an object-centric dataset that contains no information on the background diversity. Due to the nature of ImageNet-1K, our paper focuses on object-level covariate shifts, which is an orthogonal issue to background-level covariate shifts.
>
> Regarding semantic shift, this is an excellent suggestion and we updated the paper to include semantic shift results with ImageNet-OOD in Figure 4 (page 7). The results revealed that OOD detection algorithms on randomly-initialized models do perform very close to random chance when evaluated with the semantic shift of ImageNet-OOD. This is in contrast to covariate shift shown with ImageNet-R or ImageNet-C, which the OOD detector has a bias either towards detecting as ID or OOD.

---

> > ### Comment · Reviewer_JbD1 · 2023-11-21
> > **Final score**
> >
> > Thank you for the detailed response. I have raised my score to 6 recommending for this paper to be accepted.

---

### Meta-Review · Area_Chair_mi1v · 2023-12-05

**Metareview:**

This paper introduces ImageNet-OOD, a new out-of-distribution (OOD) detection dataset that minimizes covariate shift compared to ImageNet-1K by manually selecting semantically different classes from ImageNet-21K. Using this dataset, the authors evaluate several recent OOD detection algorithms and find they offer little benefit over the maximum softmax probability (MSP) baseline for detecting semantic shifts. The key conclusions are several folds. Modern OOD detectors are much more sensitive to covariate shifts than semantic shifts. On ImageNet-OOD, modern OOD detectors offer <1% AUROC improvement over MSP for new-class detection. For failure detection, MSP still outperforms modern OOD detectors on ImageNet-OOD. The benefits of modern OOD detectors come more from better separating incorrect in-distribution examples rather than improving on semantic shifts.

The clarity and vision are above the bar of ICLR. While the reviewers had some concerns on the related works and comparison methods, the authors did a particularly good job in their rebuttal. Thus, all of us have agreed to accept this paper for publication! Please include the additional experimental results and discussion in the next version.

**Justification For Why Not Higher Score:**

The clarity and vision are above the bar of ICLR. While the reviewers had some concerns on the related works and comparison methods, the authors did a particularly good job in their rebuttal. Thus, all of us have agreed to accept this paper for publication! Please include the additional experimental results and discussion in the next version.

**Justification For Why Not Lower Score:**

The clarity and vision are above the bar of ICLR. While the reviewers had some concerns on the related works and comparison methods, the authors did a particularly good job in their rebuttal. Thus, all of us have agreed to accept this paper for publication! Please include the additional experimental results and discussion in the next version.

---

### Decision · Program_Chairs · 2024-01-16

Accept (poster)